

# Modeling Mediterranean ocean biogeochemistry of the Last Glacial Maximum

Katharina D. Six[1], Uwe Mikolajewicz[1], and Gerhard Schmiedl[2]

[1]Max Planck Institute for Meteorology, Hamburg, Germany
[2]University Hamburg, Hamburg, Germany

**Correspondence:** Katharina D. Six (katharina.six@mpimet.mpg.de)

**Abstract.** We present results of simulations with a physical-biogeochemical ocean model of the Mediterranean Sea for the last glacial maximum (LGM) and analyse the difference in physical and biochemical states between the present day and the past. Long-term simulations with an Earth system model based on ice sheet reconstructions provide the necessary atmospheric forcing data, oceanic boundary conditions at the entrance to the Mediterranean Sea, and river discharge to the entire basin. Our
regional model accounts for changes in bathymetry due to ice-sheet volume changes, reduction in atmospheric carbon content, and an adjusted aeolian dust and iron deposition. The physical ocean state of the Mediterranean during the LGM shows a reduced baroclinic water exchange at the Strait of Gibraltar, a more sluggish zonal overturning circulation, and the relocation of intermediate and deep water formation areas - all in line with estimates from paleo sediment records or previous modelling efforts. Most striking features of the biogeochemical realm are a reduction of net primary production, an accumulation of
nutrients below the euphotic zone, and an increase of organic matter deposition at the sea floor. This seeming contradiction of increased organic matter deposition and decreased net primary production challenges our view of possible changes in surface biological processes during the LGM. We attribute the origin of a reduced net primary production to the interplay of increased stability of the upper water column, changed zonal water transport at intermediate depths, and colder water temperatures, which slow down all biological processes during the LGM. The cold water temperatures also affect the remineralisation rates
of organic material which explains the simulated increase of organic matter deposition, in good agreement with sediment proxy records. In addition, we discuss changes of an artificial tracer which captures the surface ocean temperature signal during organic matter production. A shifted seasonality of biological production in the LGM leads to a difference in the recording of the climate signal by this artificial tracer of up to 1 K. This could be of relevance for the interpretation of proxy records like e.g. alkenones. Our study does not only provide the first consistent insights into the biogeochemistry of the glacial Mediterranean
Sea, it will also serve as the starting point for long-term simulations over the entire last deglaciation.



# 1 Introduction

The Earth climate during the Last glacial maximum (LGM) is characterized by much colder climate conditions and the existence of massive ice sheets on the northern hemisphere which led to a lowering of the sea level of 70-130 m (Lambeck et al.,
2014; Peltier et al., 2015; Tarasov et al., 2012) and significant changes in the atmospheric and oceanic circulation (Löfverström and Lora, 2017) including changes in precipitation and temperature patterns (Kageyama et al., 2021; Löfverström, 2020). These global variations also affect the physical and biogeochemical states of the semi-enclosed Mediterranean Sea (MedSea), which is connected to the global ocean at Gibraltar. Today's typical anti-estuarine thermohaline overturning circulation (ZOC) with less saline, and nutrient poor water entering at Gibraltar and a salty and nutrient-enriched deeper return flow of primarily
Levantine intermediate water masses was slowed down during the LGM (Mikolajewicz, 2011; Colin et al., 2021). Modelling studies indicated that the primary causes of the ZOC variation are the reduced through-flow area at Gibraltar at low sea-level stands (Mikolajewicz, 2011; Colin et al., 2021) and an increased water column stability hampering the formation of Levantine intermediate water (Rohling, 1991; Mikolajewicz, 2011). Proxy data from foraminifera shells confirm reduced water exchange during the LGM between the eastern and the western MedSea basins (Duhamel et al., 2020; Cornuault et al., 2018) and point
towards a strong salinity increase in the eastern basin (Thunell and Williams, 1989). The globally more arid LGM climate would support higher salinities, but colder conditions also reduce evaporative fluxes. The net water loss of the MedSea for the LGM is approximately only half of that for modern times (Mikolajewicz, 2011). Tendencies of precipitation and river runoff at the LGM are rather uncertain. A multi-model analysis of paleo-simulations with Earth system models (Kageyama et al., 2021) showed a local precipitation increase over the MedSea, which is also supported by the study of Goldsmith et al. (2017) on
long-term water isotopic composition over Israel. In contrast, pollen records concordantly suggest expansion of steppe vegetation and thus significant aridification during the LGM (Allen et al., 1999; Kotthoff et al., 2008; Koutsodendris et al., 2023) and sediment core data which capture climate changes in the Nile catchment rather indicate arid conditions and reduced river discharge compared to present (Box et al., 2011; Revel et al., 2014).

While there are a small number of modelling attempts to reconstruct the physical ocean of the MedSea during the LGM
(Myers et al., 1998; Rohling and Gieskes, 1989; Mikolajewicz, 2011), there are none at all published for glacial biogeochemical conditions of the MedSea. Thus, our understanding of tendencies in biogeochemical tracers between past and present stems solely from the interpretation of sediment core data. Lateglacial faunal records from benthic foraminifers indicate that organic matter flux to the seafloor during the LGM was significantly higher than today (Kuhnt et al., 2007; Abu-Zied et al., 2008; Schmiedl et al., 2010). Commonly, it is widely assumed that organic matter fluxes to the seafloor in oligotrophic regions without
oxygen limitations are highly correlated to surface primary production (Betzer et al., 1984). As an example, for the present day MedSea, the decreasing primary production from the western to the eastern basin and the concurrent resulting decrease in food availability at the seafloor is clearly reflected in the benthic foraminiferal composition (Rijk et al., 2000). Consequentially, an enhanced organic matter flux during the cold period would point to a higher primary production, quantified, for example, for a single sediment core in the western MedSea (Alboran Sea) which shows a 10 % increase in annual primary production (Radi
and de Vernal, 2008). Benthic foraminiferal stable carbon isotope records suggest approximately 20 % higher glacial primary



productivity in the Ligurian Sea but no significant changes in the Strait of Sicily (Theodor, 2016).

Increased LGM primary production due to upward mixing of nutrients into the euphotic layer was also postulated by Rohling and coworkers based on theoretical considerations about a shallowing of the pycnocline position in the eastern basin (Rohling and Gieskes, 1989; Rohling, 1991). However, changes in surface nutrient supply during the LGM are still unclear, which
impedes also the interpretation of sediment core data.

We present results of a new model framework (medHAMOCC) to investigate the physical and biogeochemical conditions in the MedSea during the LGM. The setup consists of a regional ocean-biogeochemical model for the MedSea which is forced by unique new data sets from an Earth system model (MPIESM) suitable for long-term paleo simulations (Kapsch et al., 2022). The MPIESM provides all necessary atmospheric and lateral oceanic physical boundary conditions and river runoff from the
last glacial to present day (26,000 years) for two different ice-sheet reconstructions (see section 2.2). With these consistent data sets we investigate the mean state of the MedSea for two time slices: the LGM and present day. We also consider changes in bathymetry and in nutrient river loads for the LGM. The efficient performance of the regional MedSea model allows for a 1000 years spin-up run followed by a 1000 years simulation for each time slice. This guarantees a very low model drift and also minimises the impact of initial conditions. Following a model evaluation of present-day conditions (section 3), we address the
drivers of circulation changes between past and present and analyse in detail their impact on the biogeochemistry (section 4). An additional sensitivity run is performed to gain further insights into the impact of a changed LGM bathymetry (section 2.3). By introducing a diagnostic tracer to track biological production temperatures, we mimic the information captured in paleo-proxy data records. This gives us the opportunity to investigate potential biases that may occur when recording the climate signal in marine sediment cores (subsection 4.4). Future applications and limitations of our model framework will be discussed
at the end of this paper (section 5).

## 2  Methods

### 2.1  Model setup

We use a regional setup of the primitive equation ocean general circulation model MPIOM (Mikolajewicz, 2011; Jungclaus et al., 2013; Mauritsen et al., 2019) combined with a comprehensive biogeochemical model HAMOCC (Six et al., 1996; Ilyina
et al., 2013) in an updated version of the one used by Liu et al. (2021). This setup is called medHAMOCC in the following. MPIOM is formulated on an Arakawa-C grid in the horizontal and on z-levels in the vertical direction using the hydrostatic and Boussinesq approximations. Subgrid-scale parameterizations include lateral mixing on isopycnals (Redi, 1982) and tracer transports by unresolved eddies (Gent et al., 1995). Vertical mixing is realized by a combination of the Richardson number-dependent scheme of Pacanowski and Philander (1981) and direct wind-driven turbulent mixing in the mixed layer (further
details are given in Jungclaus et al. (2013) and Mauritsen et al. (2019)). The Mediterranean setup of the MPIOM covers the entire Mediterranean Sea and the Black Sea. Part of the Atlantic is included as a sponge zone for the open western boundary (Fig. 1). The numerical grid has a mean horizontal resolution of approx. 20 km and 42 unevenly distributed layers in the vertical (9 layers in the upper 100 m and increasing layers thicknesses to a maximum and constant layer thickness of 150 m below 675



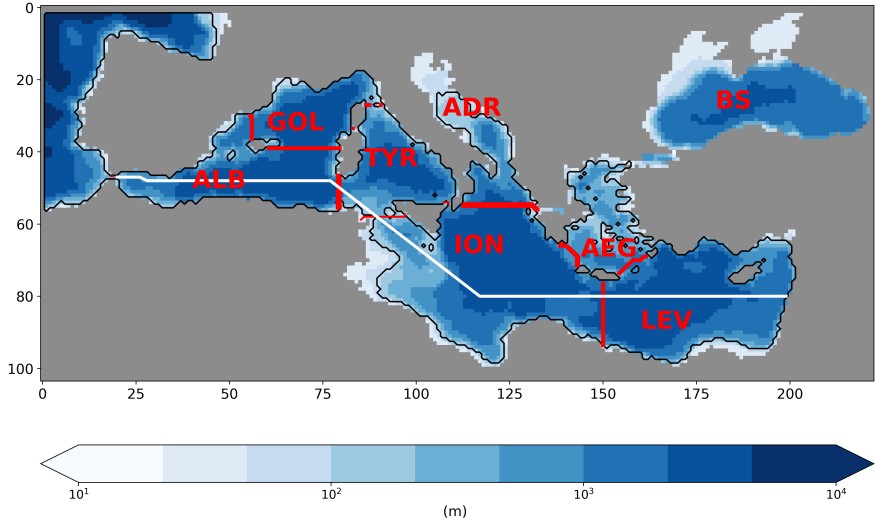

**Figure 1.** Topography of present day (depth in m) for the entire model domain (225x104 grid cells). LGM shoreline is given by the black contour line. Red lines indicate extension of regions used for basin averages: ALB=Alboran, GOL=Gulf of Lions, TYR=Tyrrhenian Sea, ION=Ionian Sea, LEV=Levantine Sea, AEG=Aegean Sea, ADR=Adriatic, BS=Black Sea. The white line shows the geographical location of the west-east section.

m). The bathymetry for present day and of the Last Glacial Maximum is derived from ice-sheet reconstructions following the

method of Meccia and Mikolajewicz (2018). We consider partial grid cells in the last wet grid cell. Fig. 1 shows the present day land-sea mask and basin depth. The overlaid LGM shoreline indicates a closed Bosporus gateway and the desiccation of the northern Adriatic Sea due to low sea level stands (see section 2.3). The same hydrodynamic processes that are resolved for tracers within MPIOM are applied to all tracers of the biogeochemical component HAMOCC (Six et al., 1996; Ilyina et al., 2013; Paulsen et al., 2017). HAMOCC includes a description of the full carbon chemistry, plankton dynamics for phytoplank-

ton, zooplankton and nitrogen fixers, and dissolved and particulate organic matter (the latter called detritus in the following). HAMOCC includes detritus settling and remineralization (aerobic and anaerobic), and production, sinking and dissolution of opal and calcite shells (Ilyina et al., 2013; Mauritsen et al., 2019). Organic matter produced by primary production is composed of carbon, phosphorus, nitrogen, and oxygen according to a constant stoichiometry (C:N:P:$O_2$=122:16:1:-172, Takahashi et al. (1985)) and iron (Fe:P=366 $10^{-6}$, Johnson et al. (1997)). A sediment module, which consists of solid components (detritus,

opal and calcite shells, clay, and sand), includes the same remineralisation and dissolution processes in porewater as in the water column and acts as the lower boundary (Heinze et al., 1999). The sediment module has the same horizontal resolution as med-HAMOCC and 12 biologically active levels with increasing layer thickness and decreasing porosity from the water-sediment interface to a diagenetically consolidated burial layer.

Two major changes have been introduced in the here applied medHAMOCC compared to the latest version of HAMOCC6

(Mauritsen et al., 2019) :

1) a modified sinking and remineralization process of detritus: in general, gravitational driven detritus fluxes show the shape





of a power law function (Martin et al., 1987). Variations in the settling velocity, which are attributed to disaggregation and aggregation of particles or changes in the intensity of remineralisation, e.g. due to temperature dependent bacterial activity, form this curve (Kriest and Oschlies, 2008; Laufkötter et al., 2017). As the exponent of the power law function is given by the

ratio of the remineralization rate and the particle settling velocity, a decrease in the remineralisation rate has the same effect on the exponent as an increase in the settling velocity (Kriest and Oschlies, 2008). In HAMOCC6, a depth-dependent detritus sinking speed and a temperature-independent remineralization rate are applied. In view of the large temperature difference between LGM and present day, we introduce a temperature-dependent remineralisation rate (Bidle et al., 2002) and a constant detritus sinking speed in medHAMOCC. The control parameters were optimized to obtain an equivalent result between the

two parametrisation approaches for present day. Note, that the remineralisation rate is oxygen-dependent in both HAMOCC versions. 2) a sediment resuspension: we implemented the sediment resuspension scheme developed in Mathis et al. (2019). Depending on grain size, particle density and bottom shear stress, all solid particles (except of sand) from the upper most sediment layer can be remobilized and are relocated or remineralized in the water column.

Additionally, we introduce a tracer ($T_{org}$) to track the temperature at which primary production takes place. $T_{org}$, a kind of a

phytoplankton twin, is treated by the same biological turnover processes and eventually enters the sediment by organic matter settling. It mimics the information which is implicitly carried by proxy data, such as alkenones. $T_{org}$ gives us the possibility to assess potential biases that may occur when climate signals are recorded by proxy data.

## 2.2 Forcing and boundary conditions

All necessary atmospheric forcing fields are taken from long-term paleo simulations with an Earth system model (MPIESM,

Kapsch et al. (2022)). These transient simulations cover the entire period from the last glacial to present day (26,000 years), but we use here only two time slices : the LGM and the preindustrial period (PI). To allow for a consistent spin-up, we take 2000 years transient forcing data for each time slice: LGM (22000-20000 yr BP) and PI (2000-0 yr BP, reference year 1950) and we consider the first 1000 years of each time slice simulation as spinup. Atmospheric forcing data are 2-meter temperature, dew point temperature, downward short and long wave radiation, 10 m wind speed, zonal and meridional wind stress components,

and precipitation as monthly averages. The atmospheric model component of MPIESM has a relative coarse horizontal resolution (approx. 3.75 °). Therefore, we applied a simple downscaling procedure with bias correction on the basis of reanalysis data (ERA 20C, Poli et al. (2016)). Heat and freshwater fluxes over the water are calculated using the standard bulk formulae (Mikolajewicz, 2011; Liu et al., 2021).

A unique feature of these long term simulations with MPIESM is an automatic adaptation of the topography/bathymetry due

to volume changes of ice sheets and isostatic adjustment which is calculated every 10 simulation years (Kapsch et al., 2022). Corresponding changes of the hydrological discharge and the river routing are also considered (Riddick et al., 2018). For medHAMOCC, we adopt the same automatic adaptation of the topography/bathymetry and consider the transient hydrological discharge to the MedSea. We add freshwater from the river runoff with a corresponding nutrient load (phosphate, nitrate and silicate). This nutrient load is deduced from simulated freshwater discharge multiplied with a mean river nutrient concentration

for individual basins of the Mediterranean Sea and the Black Sea to account for temporal variability in nutrient supply. We





derive the basin mean nutrient concentration from modern nutrient input estimates (Ludwig et al., 2009) divided by the simulated mean basin river discharge of the last 10 years of the MPIESM simulation. This procedure guarantees a realistic nutrient supply for present day despite a potential river discharge bias in the parent simulation. Aeolian clay and iron deposition for the PI and the LGM is taken from Albani et al. (2016).

At the open boundaries of the model domain towards the Atlantic Ocean, we include a sponge zone of about 80 km over which ocean temperatures and salinities are relaxed to corresponding monthly mean data from the MPIESM simulation with a relaxation time of 1 month. These boundary conditions of MPIESM have been bias-corrected by the difference between the mean of the last 100 yr of the MPIESM simulation and a climatology for temperature and salinity (PHC, Steele et al. (2001)). Sea level is relaxed to zero in the sponge zone. All biological organic components (e.g. phytoplankton) are relaxed to

their prescribed minimum concentration. For other biogeochemical tracers we use climatological monthly mean data from the World Ocean Atlas (Garcia et al. (2019a, b), phosphate, nitrate, oxygen, and silicate) and climatological annual mean data from GLODAPV2 (Lauvset et al. (2021), dissolved inorganic carbon, and alkalinity). We are aware of the inconsistency to prescribe modern concentrations in the sponge zone in a LGM simulation. However, estimates on nutrient concentrations for the LGM remain a challenge. Tamburini and Föllmi (2009) suggested an overall increase of the global phosphate inventory of 17-40 %

between LGM and present day. Some sediment records along the Iberian margins (Kohfeld et al., 2005; Radi and de Vernal, 2008) and a few modelling studies (Bopp et al., 2003; Menviel et al., 2008; Palastanga et al., 2013) point towards higher primary production in the North Atlantic as a result of enhanced nutrient upwelling and/or nutrient supply from shelf areas due to sea level low stands. However, a higher global phosphate inventory does not imply that surface nutrient concentrations are also higher in the North Atlantic (Bopp et al., 2003; Morée et al., 2021) and that more nutrients would enter the Mediterranean

Sea. Therefore, we stick to present day's values for the prescribed boundary conditions in the sponge zone.

## 2.3    Set of experiments

From MPIESM, we use forcing data sets from two simulations based on different ice sheet reconstructions (GLAC-1D, Tarasov et al. (2012) and ICE-6G, Peltier et al. (2015), see Tab. 1). Differences in these "parent" simulations arise primarily from the temporal change in ice volume and its impact on atmospheric circulation, as well as the freshwater discharge from ice melt affecting sea surface salinity and, consequentially, ocean meridional overturning circulation in the North Atlantic (Kapsch et al.,

2022). Estimates of different ice sheet volumes during the LGM for GLAC-1D and ICE-6G lead to different eustatic sea level changes of about 70 m and 100 m, respectively. Consequentially, the present day sill depth at Gibraltar (297 m) shallows to 216 m in GLAC-1D and 200 m in ICE-6G. By applying forcings from different ice sheet reconstructions we capture uncertainties resulting from glacial boundary conditions. Each parent simulation provides 2000 years of a consistent forcing for the LGM

(22000-20000 yrs BP) and preindustrial (2000-0 yrs BP with a reference year of 1950) time slices. Our results are shown as means for the last 1000 years of each simulation with an exception for the model-data comparison with the MEDATLAS (MEDAR Group, 2002) when we use only the last 70 years of the PI simulation of GLAC-1D and ICE-6G.

In addition, we perform a sensitivity study with the present-day GLAC-1D forcing to assess only the impact of shallower sill depths at Gibraltar and in the Strait of Sicily on the circulation of the MedSea. To this end, we reduce the sill depth at Gibraltar





and the Strait of Sicily to LGM values (216 m for Gibraltar and 252 m at Strait of Sicily) in experiment PI-Straits (Table 1).
The remaining bathymetry is unchanged.

| Experiment name | Climate forcing | maximum sill depths (m) | |
|---|---|---|---|
| | | Gibraltar | Strait of Silicy |
| GLAC-1D | PI with Tarasov et al. (2012) | 297 | 355 |
| | LGM with Tarasov et al. (2012) | 216 | 252 |
| ICE-6G | PI with Peltier et al. (2015) | 297 | 355 |
| | LGM with Peltier et al. (2015) | 200 | 237 |
| PI-Straits | PI with Tarasov et al. (2012) | 216 | 252 |

**Table 1.** List of experiments and their major differences.

## 3   Assessment of the physical and biogeochemical state of the PI

Before we discuss the LGM results, we briefly assess the performance of the physical and biogeochemical state for the PI
simulations. Both simulations show very similar results for temperature and salinity for the PI, which is attributable to the very
similar PI states of the parent simulations. The basin-wide average of temperature for the Gulf of Lions (GOL, Fig. 2a) agrees
quite well with the data compilation of the MEDATLAS (MEDAR Group, 2002), but model results are slightly too cold for the
Levantine basin (LEV, Fig. 2e). Both mean salinities of the basins are fresher (0.2-0.6) compared to the observations, but the
simulated surface salinity difference between GOL and LEV (0.76-0.82) is comparable to the observed one (approx. 1). Note,
that we used only the last 70 years of the PI simulations for this model-data comparison.

The MedSea is an evaporative basin where evaporation exceeds precipitation and river input (PEM = preciptation + river
runoff - evaporation). Surface salinities are the result of the interplay between the net water budget, basin water residence time
and the salinity signal of inflowing North Atlantic water. Simulated absolute PEM (69-72 1000 $m^3$ $s^{-1}$ in both PI simulations)
is slightly higher than observational estimates (22-52 1000 $m^3$ $s^{-1}$, Rohling et al. (2015) and refs within; Sanchez-Gomez
et al. (2011)) and simulated river runoff to the entire MedSea including the Black Sea contribution at the Bosporus (in total
about 5-6 1000 $m^3$ $s^{-1}$) is much lower than the observational estimates (Dubois et al. (2011): 17.5 1000 $m^3$ $s^{-1}$, Ludwig et al.
(2009): 13.6-23.3 1000 $m^3$ $s^{-1}$) (Table 2). Our atmospheric parent model, which has a very coarse horizontal resolution (3.75
°), potentially underestimates the total runoff to the entire region as already shown for MPI-ESM model versions with even
higher spatial resolution (Sanchez-Gomez et al., 2011). The net water loss through evaporation is compensated by a net in-
flow of Atlantic water (+0.06-0.076 Sv) which is within the range of observational estimates ( +0.04-0.11 Sv, Candela (2001);
Tsimplis and Bryden (2000) ).  Figure 3 shows the individual in- and outflow contributions for PI which are with 1.15-1.25
Sv at the upper end of range of estimates (e.g. for the outflow, Tsimplis and Bryden (2000) : 0.78 Sv, Candela (2001) : 0.97
Sv, Lacombe and Richez (1982) : 1.15 Sv). The simulated net heat budget (-6 to -4 W $m^{-2}$) compares well to observational

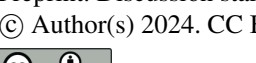



|  | OBS | GLAC-1D | | ICE-6G | |
| --- | --- | --- | --- | --- | --- |
|  |  | PI | LGM | PI | LGM |
| precipitation | 20.3±3.5 to 47.1±4.4 | 40.5 | 39.7 | 38.5 | 40.7 |
| evaporation | -90.1±7.1 to -86.8±4.7 | -110.2 | -84.6 | -110.7 | -88.5 |
| river runoff | 11.2±1.7 (+6.3±3.5) | 4.53(+1.6) | 5.76 | 4.35(+1.0) | 6.39 |
| net flux | -52.±8.7 to -22.2±8.7 | -65.1 | -39.1 | -67.8 | -41.4 |

**Table 2.** Hydrological bugdet of the Mediterranean Sea for present day from observations (OBS,Sanchez-Gomez et al. (2011)) and of PI and LGM for GLAC-1D and ICE-6G. Numbers given in 1000 m$^3$ s$^{-1}$. For OBS, numbers are converted from mm yr$^{-1}$ with an area for the Mediterranean of 2.5 10$^{12}$ m$^2$. Numbers in brackets for river runoff gives the Black Sea contribution for PI. The Bosporus gateway is closed during the LGM.

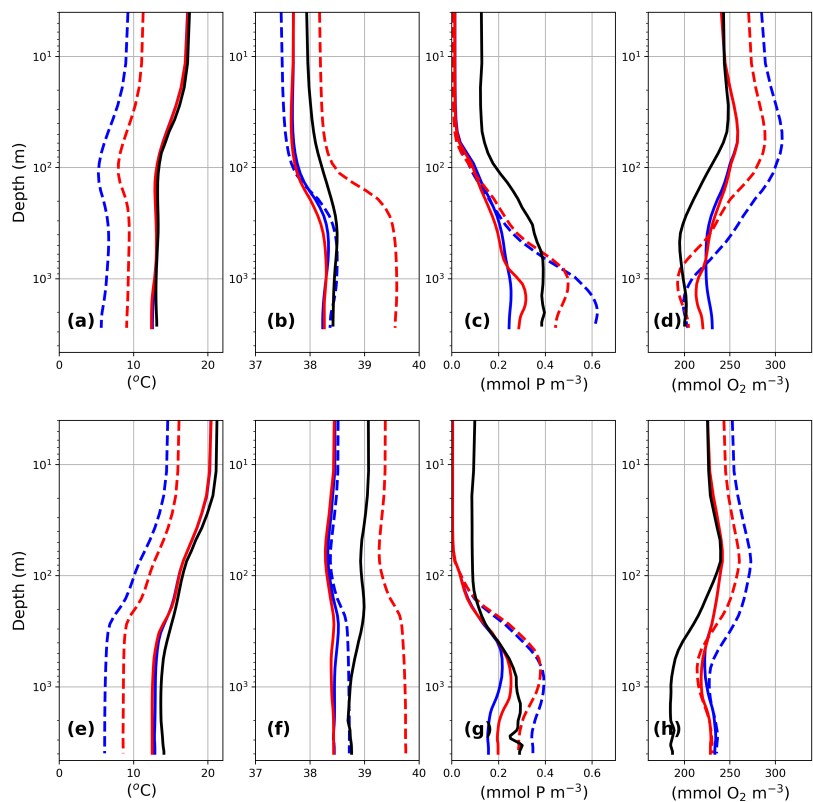

**Figure 2.** Basin-wide averages for temperature (a, e, °C), salinity (b, f), phosphate (c, g, mmol/m$^3$), and oxygen (d, h, mmol/m$^3$) for the Gulf of Lions (GOL, a-d) and the Levantine (LEV, e-h) for GLAC-1D (blue) and ICE-6G (red). Solid lines refer to the PI, dashed lines to the LGM. The black line indicates basin-wide averages of observations (MEDAR Group, 2002). For this comparison to observations we averaged only over the last 70 years of the PI simulations. Note the logarithmic depth axes. See Fig. 1 for basin definitions.




| | OBS | GLAC-1D | | ICE-6G | |
|---|---|---|---|---|---|
| | | PI | LGM | PI | LGM |
| absorbed shortwave rad. | 187±3 | 198.9 | 204.1 | 199.1 | 206.9 |
| net longwave radiation | -84±1 | -75.2 | -81. | -75.5 | -82.6 |
| latent heat flux | -90±7 | -112.3 | -97.4 | -112.7 | -104.5 |
| sensible heat flux | -14±2 | -16.4 | -31.8 | -16.5 | -24.7 |
| total | -1±8 | -5.0 | -6.0 | -5.6 | -4.8 |

**Table 3.** Annual mean heat fluxes of the Mediterranean Sea (W m$^{-2}$) from observations (OBS, Sanchez-Gomez et al. (2011)) and of PI and LGM for GLAC-1D and ICE-6G

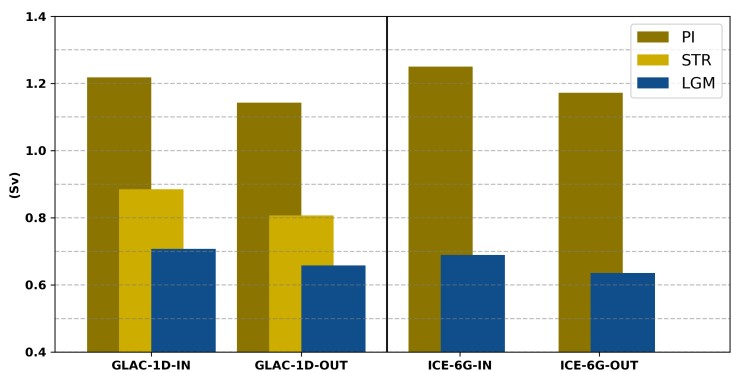

**Figure 3.** Baroclinic water transport (Sv) at Gibraltar separated into in- and outflow components for the PI (brown bars) and the LGM (blue bars) of GLAC-1D and ICE-6G. Golden bars indicate the results of PI-Straits (STR). See also legend in the upper right corner.

estimates (-7 to -1 W m$^{-2}$, Dubois et al. (2011)) (Table 3).

The simulated thermohaline circulation shows the characteristic clockwise zonal overturning pattern between 5 °W and 25 °E in the upper 1000 m with a maximum strength of more than 1 Sv for the PI (Fig. 4) which is in line with data estimates from Pinardi et al. (2019) or other regional modelling studies (Mikolajewicz, 2011; Adloff et al., 2011; Sevault et al., 2014). The second typical feature is a counter clockwise circulation in the deep eastern MedSea between 15 °E and 32 °E which indicates deep water formation. These intermediate and deep water formation areas in the PI are located in the Gulf of Lions, the southern Adriatic Sea, and the Aegean Sea which is indicated by the maximum annual mixed layer depth (Fig. A1). The maximum strength of this deep counter clockwise cell in the PI (0.08-0.095 Sv) is slightly weaker than estimates from other modeling studies (Mikolajewicz, 2011; Adloff et al., 2011; Sevault et al., 2014), which is a consequence of the reduced variability of the applied monthly mean forcing and the long averaging period (here over 1000 years).





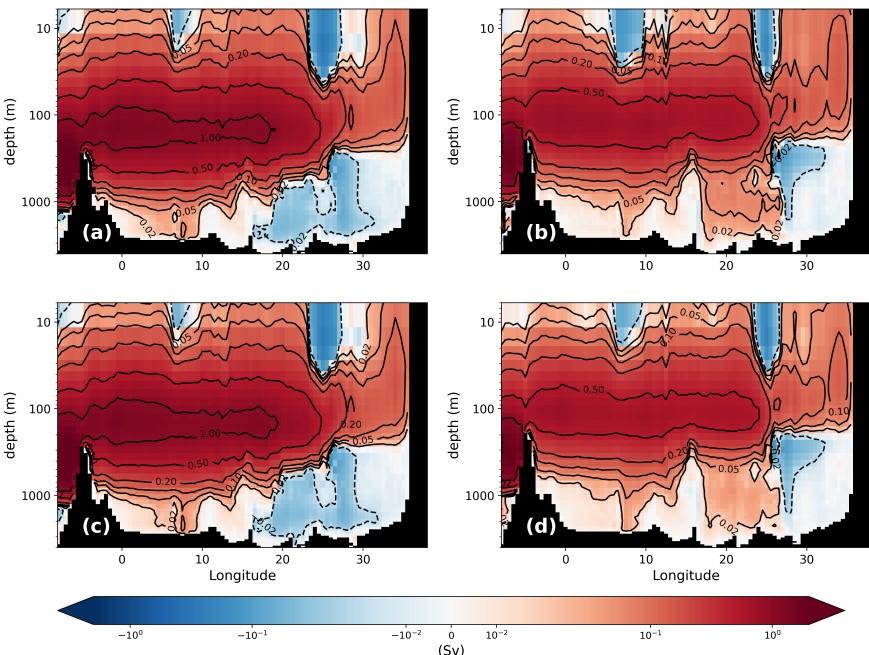

**Figure 4.** Mean zonal stream function (Sv) of GLAC-1D (a, b) and ICE-6G (c, d) for the PI (a, c) and the LGM (b,d). Contour lines are given as dashed line for (-0.02), and as solid lines for (0.02), (0.05), (0.1), (0.2), (0.5), and (1.0). Please note the logarithmic depth scale and the logarithmic color scale .

All simulated biogeochemical patterns of the PI reflect the typical west-east gradient. Higher nutrient surface concentrations
and a correspondingly higher net primary production (NPP) are found in the western basin, while more oligotrophic conditions
are simulated for the eastern basin, with the exception of the region downstream of the Nile river mouth (Fig. 5a). Note, that
our simulation is based on conditions prior Nile damming. High NPP is found in the Alboran Sea and in the modified Atlantic
water flowing along the coast of Africa into the Ionian Sea. A local high is also found in the GOL fueled by nutrient supply
due to deep winter mixed layers. The oligotrophic eastern Levantine has the lowest net primary production, about half that of
the GOL, which is in line with observational estimates (e.g. Uitz et al. (2012)).

Basin-wide mean phosphate profiles below 100 m fit well to observations (Fig. 2c, g) with sightly lower concentrations below
1000 m. Simulated mean surface concentrations in the euphotic zone (upper 100 m) are rather low compared to MEDATLAS
in all basins. However, revisiting the climatological mean concentrations of the upper 150 m for 1981-2017 by Belgacem et al.
(2021) showed that the climatology of the MEDATLAS has the tendency to overestimate the nutrient concentration, e.g. in the
Gulf of Lions. Oxygen concentration below the euphotic zone are slightly higher than the climatology (20-40 mmol/m$^3$). Part
of this discrepancy could be related to the missing anthropogenic nutrient input (riverrine and atmospheric) which increased
dramatically after 1950 for N and P (Krom et al., 2014). Other nutrient sources like the simulated N-fixation show also higher
values in the western basin than in the eastern basin, which is in line with observations (Krom et al., 2014), but they play overall





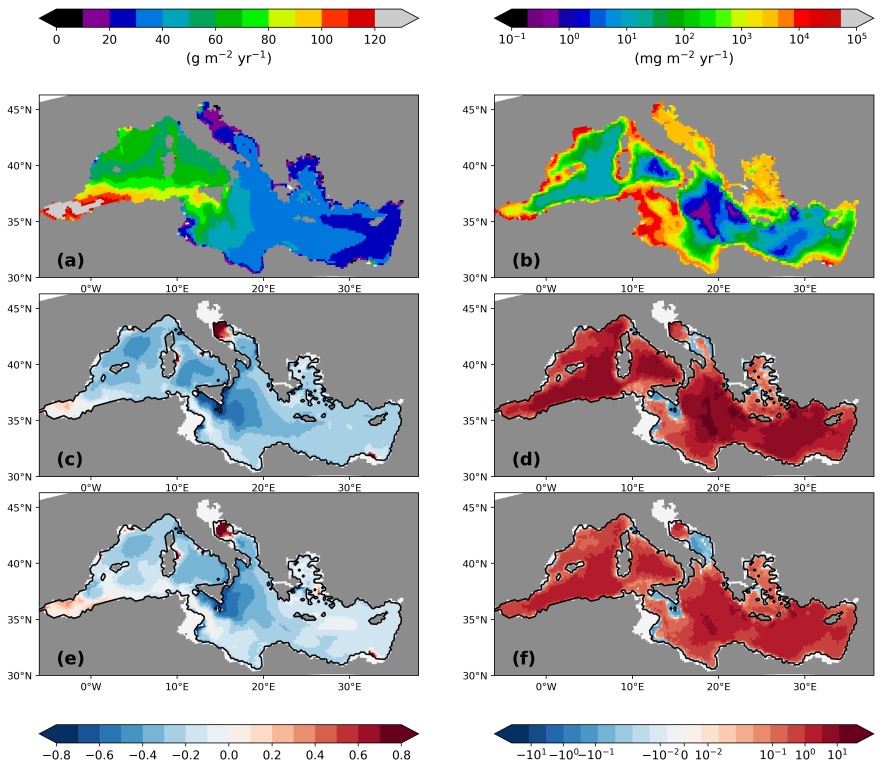

**Figure 5.** Vertical integrated net primary production NPP (a, gC m$^{-2}$ yr$^{-1}$) and settling flux of organic material into the sediment (b, mgC m$^{-2}$ yr$^{-1}$) for the PI of GLAC-1D and the relative changes between the LGM and the corresponding PI for NPP of GLAC-1D (c) and ICE-6G (e) and for the sedimentation rates of GLAC-1D (d) and ICE-6G (f). Note the logarithmic color scales in (b), (d), and (f). The black contour line indicates the LGM coastline.

a negligible role (not shown).

Biogeochemical settling processes do not only shape the vertical profiles, but also leave their imprint in the sediment. The spatial pattern of the sedimentation rate of organic material reflects primarily the local ocean depth (Fig. 5b). In the deep basins of the eastern MedSea, the sedimentation flux is less than 1 mgC m$^{-2}$ yr$^{-1}$ compared to a vertical integrated net primary production of approximately 40 gC m$^{-2}$ yr$^{-1}$. Only the shallower shelf areas receive higher organic matter fluxes as well as a higher deposition fluxes of opal shells, which is clearly visible in the weight-percentage contributions of detritus and

opal in the upper 6 mm of the sediment column (first two layers) (Fig. 6). Note, that the remaining percentage part consists primarily of calcium carbonate (CaCO$_3$) and terrigenous material (Fig. A2 ). The largest weight-percentage contribution of 40-70 % provides CaCO$_3$, where the pattern is very similar to the pattern of NPP. Similar carbonate contents have been documented in sediment cores from different Mediterranean basins (Aksu et al., 1995; Hoogakker et al., 2004; Hamann et al., 2008). Supersaturation of the entire water column in the MedSea with respect to calcite prevents dissolution of CaCO$_3$

shells (Béjard et al., 2023) and maps the pattern of planktic calcite shell production on the seabed. For detritus and opal shells,





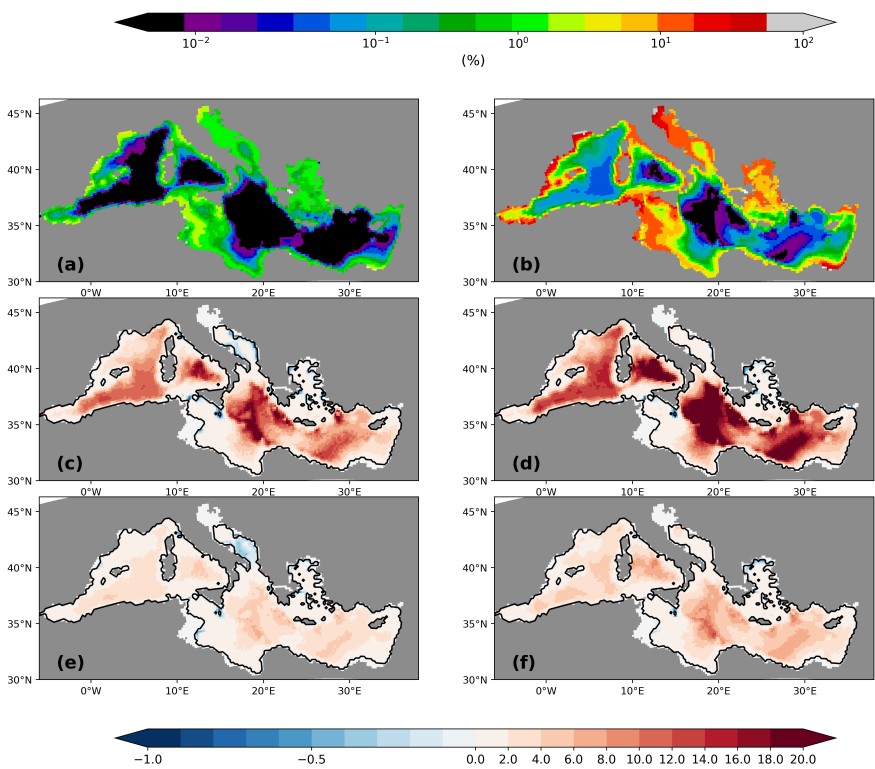

**Figure 6.** Weight-percentage sediment contribution in the upper 6 mm of the sediment column of detritus (a) and opal (b) for the PI of GLAC-1D (in %). Relative changes between LGM and the corresponding PI are given for detritus (c, e) and opal (d, f) for GLAC-1D (c, d) and ICE-6G (e, f). Note the logarithmic color scale for panels (a, b) above and the nonlinear color scale for (c, d, e, f) below the figure.

remineralization and dissolution reshape the particle settling fluxes, which decrease with increasing water depth. The sediment in all deep basins consists to less than 0.01 % of detritus or opal shells. Only in the shallower shelf areas 1-2 % weight-percent of the sediment are detritus and 2-10 % are opal shells. Given the simplicity of the sediment module and the coarse horizontal resolution of the medHAMOCC, these results fit very well of the few available core top data for present day (Pedrosa-Pàmies 240 et al., 2015). Sediment erosion, i.e. resuspension of all solid sediment material with the exception of sand, occurs primarily in the Straits of Gibraltar, the Straits of Sicily, and near shore on the Tunisian shelf. However, the simulated resuspension is probably underestimated due to the reduced variability of monthly mean forcing data, but observational data are missing to evaluate our model results.





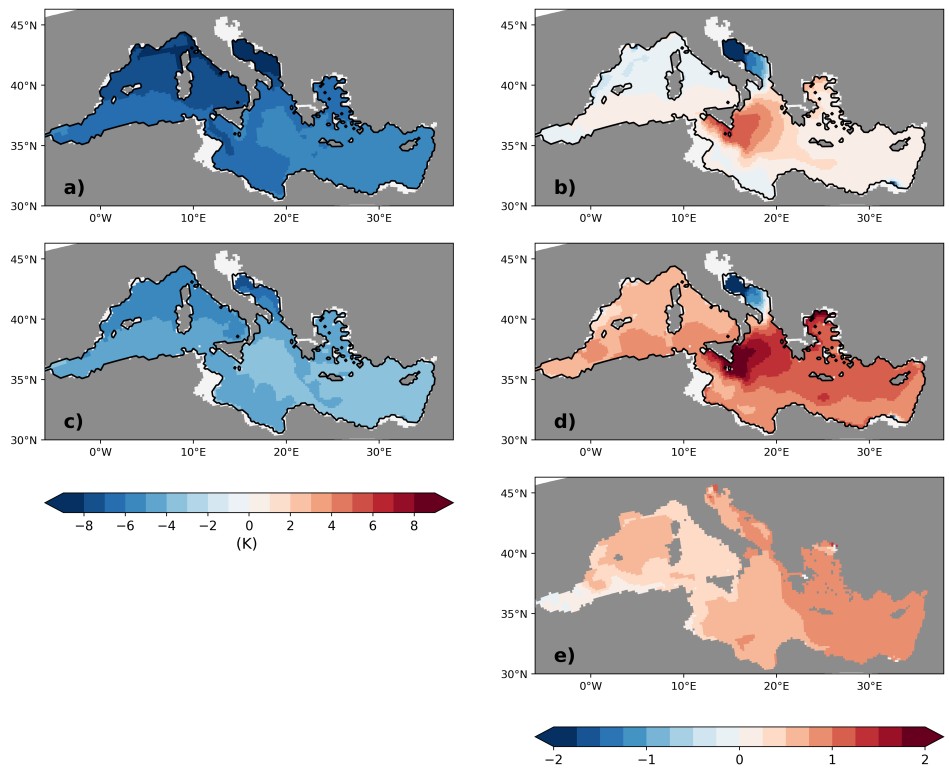

**Figure 7.** Anomalies of sea surface temperature (K) (a, c) and sea surface salinity (b, d) between LGM and PI for GLAC-1D (a, b) and ICE-6G (c, d). Panel (e) shows the anomaly of sea surface salinity between PI-Straits and PI of GLAC-1D. The anomaly of sea surface temperature between PI-Straits and PI of GLAC-1D is small (not shown).

## 4 Changes in the patterns of the Last Glacial Maximum

### 4.1 Changes of the physical environment

The sea surface temperatures of the LGM in both simulations are generally about 4-8 K lower than for the PI (Fig. 7a, c) which is in line with paleo-temperature estimates (Hayes et al., 2005; Kuhlemann et al., 2008). The comparison to SST reconstructions from planktic formaminiferal assemblages (Hayes et al., 2005) shows that model biases are within the range of results for different reconstruction methods (Fig. 8, Fig. 9). Out of the parent simulations, GLAC-1D has a weaker Atlantic overturning circulation and a relatively colder climate over the North Atlantic than ICE-6G (Kapsch et al., 2022) which is reflected in the larger SST anomaly (and larger deviation from the reconstructions, Fig. 9) throughout the entire MedSea.

For the Gulf of Lions, the annual SST of the LGM is much higher in both simulations than in the reconstruction creating a larger deviation to the proxy data compared to the eastern basin (Fig. 9). As the coarse resolution atmospheric parent models are missing the permanent ice cover over the Alps, the transport of cold air towards the gulf is potentially underrepresented in the summer season and causes this bias (Fig. A3). We also find an intensification of the SST gradient between the eastern and



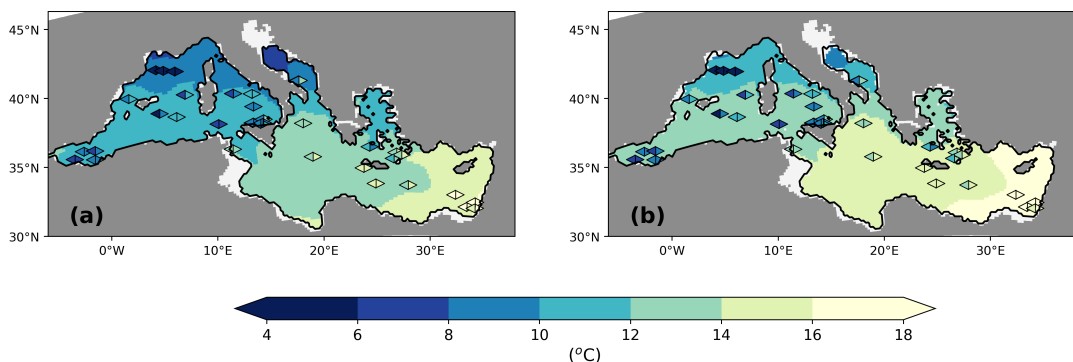

**Figure 8.** Annual mean sea surface temperature (°C) at the LGM for GLAC-1D (a) and ICE-6G (b). Overlaid data from Hayes et al. (2005) are estimates from two different methods: artificial neural network (left part of rhombus) and the revised analogue method (right part of rhombus). See Hayes et al. (2005) for more details.

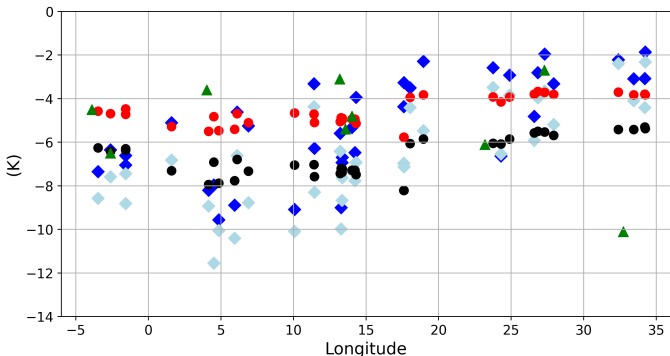

**Figure 9.** Annual mean SST differences between LGM and PI for GLAC-1D (black circles) and ICE-6G (red circles) against the longitude. Overlaid are the same SST data as in Fig. 8 using an artificial neural network (blue diamond) or the revised analogue method (light blue diamond), but here as deviation from modern SST (World Ocean Atlas, Locarnini et al. (2018)). In addition, SST change estimates based on alkenones from Lee (2004) are given (green triangle). Units are in K.



western basin during the LGM as proposed by Hayes et al. (2005). Annual mean temperature differences between GOL and LEV rises from 2 K to about 4.5 K (Fig. 2).

Sea surface salinity (SSS) changes show a very similar spatial pattern for both simulations with a higher SSS change in the eastern basin. GLAC-1D has a lower SSS increase than ICE-6G (Fig. 7b, d). This is partly due to a negative SSS anomaly at

the Gibraltar entrance (Fig. 7d). The water mass boundary of the subpolar gyre in the North Atlantic of the parent simulation of GLAC-1D shifts further to the south compared to ICE-6G during the LGM. Thus, in a colder GLAC-1D climate less saline subpolar water enters the Gulf of Cadiz and imprints on the SSS anomaly of the MedSea. The most pronounced increase in SSS is found in the Ionian Sea which is attributable to changes in the circulation linked to the freshening of the Adriatic Sea and the consequential strong reduction of local deep water formation (see Fig. A1 for changes in the maximum mixed layer

depth as indicator for deep water production). The modelling study by Mikolajewicz (2011) demonstrated a similar shift of the area of deep water formation from the Adriatic Sea to the Ionian Sea also accompanied by the largest SSS increase from present day to the LGM. In contrast to Mikolajewicz (2011), the Adriatic Sea keeps a weak anti-estuarine circulation during the LGM in our simulations.

SSS changes between LGM and present day based on $\delta^{18}$O reconstructions from planktic foraminifera were estimated to

+1.2 (+2.7) in the western (eastern) basin (Thunell and Williams, 1989) which is about 1-1.5 higher than our results. However, these estimates come with an unclear uncertainty. The underlying method combines global $\delta^{18}$O changes due to ice sheet growth (1.2 $°/_{\circ\circ}$ with a potential error of approx. 30 %, Schrag et al. (2002)), LGM temperature estimates from alkenone ratios (uncertainty $\pm 1.3\,°\mathrm{C}^{-1}$, Sijinkumar et al. (2016)), and a global $\delta^{18}$O-salinity slope regression ( 0.41 $°/_{\circ\circ}$ psu$^{-1}$, Thunell and Williams (1989), 0.25-0.45 $°/_{\circ\circ}$ psu$^{-1}$, Emeis et al. (2000)). Sijinkumar et al. (2016) performed an error propagation for

reconstructed LGM salinities in the Bay of Bengal and calculated an error range of 1.8-2.6 for salinity change estimates of 0.9-3.5 between LGM and the modern ocean. Given these large uncertainties in the SSS reconstruction, we cannot assess our model results based on these literature estimates. Our simulated SSS changes are at least consistent with the applied forcing. Besides the changing salinity in the Gulf of Cadiz, which is determined by the parent simulation, we find e.g. that the spatial SSS difference between the Levantine and the Gulf of Lions increases by 0.3 from the PI to the LGM in GLAC-1D (0.5 for

ICE-6G) (Fig. 2). This is attributable to a longer residence time of the water within the basins due to a reduced zonal overturning stream function (ZOC), which also means that the surface water is exposed to net evaporative fluxes for a longer time.

The ZOC of both experiments show that the strong, deep reaching anti-estuarine circulation (max transport of 1.2 Sv, vertical extension to 600-800 m) in the PI is reduced to  0.6 Sv and reaches only to 400 m depth in the LGM (Fig. 4). The reduction of the ZOC is linked to the change of the outflow at Gibraltar due to lower sill depth (Fig. 3, Table 1), changed freshwater budget

(Table 2), and a concurrent increase of the salinity gradient between surface and intermediate layers (0.17-0.265 between 466 m and surface, Fig. 2). The magnitude of the simulated ZOC reduction fits well to the previous estimates of 40-65 % from modelling studies (Myers et al., 1998; Mikolajewicz, 2011) and theoretical considerations (Rohling (1991) and refs within). About two-third of the ZOC reduction can be attributed to a reduced cross-sectional area at the Strait of Gibraltar as demonstrated by the sensitivity study PI-Straits (Fig. 3).

To gain more insight into the combined effects of LGM changes in SST, SSS and ZOC on the circulation, we investigate




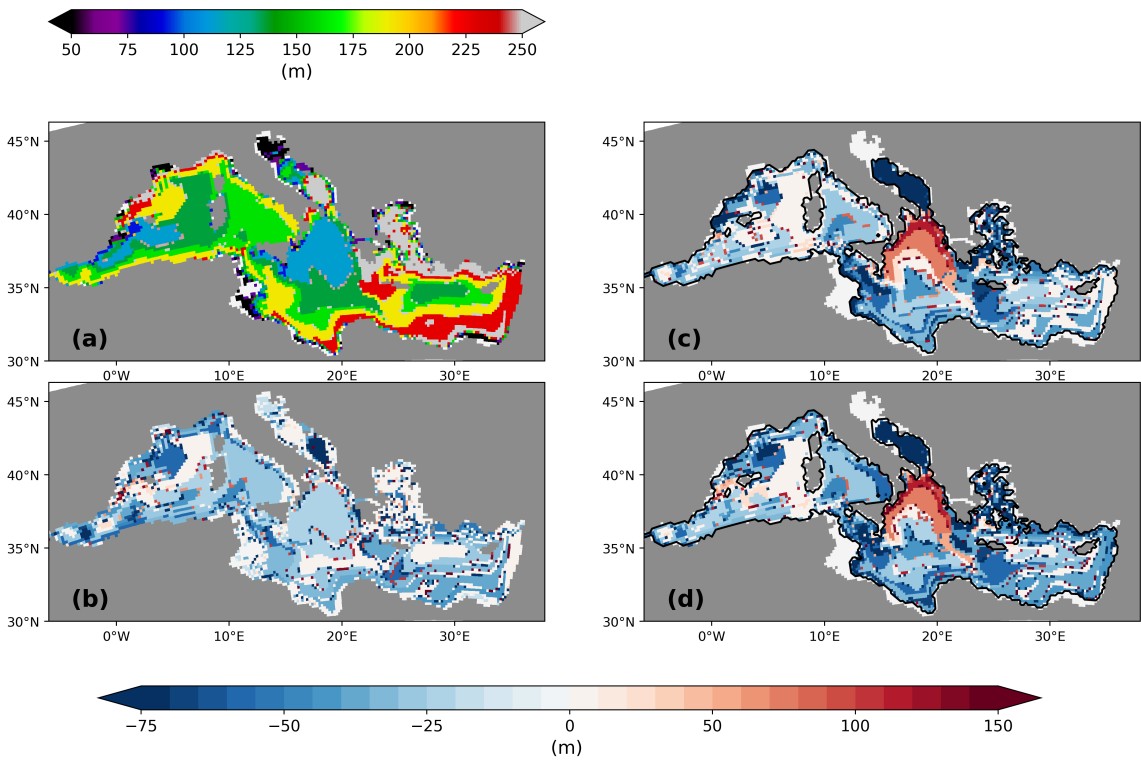

**Figure 10.** Depth of pycnocline in February for PI of GLAC-1D (a) and the pycnocline depth change between LGM and PI for GLAC-1D(b) and ICE-6G (d). Panel (b) gives the pycnocline depth change between PI-Straits and the PI of GLAC-1D. Units are in (m).

the depth change of the pycnocline in February, when a shallower winter pycnocline depth could allow for a higher nutrient supply to the productive surface layers as postulated by Rohling (1991). The pycnocline depth in February for the PI of both experiments is very similar as a consequence of the similarity of the parent simulations for PI. Fig. 10 shows, therefore, only the result of the PI for GLAC-1D. For the LGM, we see a shallowing of the pycnocline by 20-60 m for most areas of the MedSea

in both simulations. This corresponds to a relative shallowing to about 65-85 % between the LGM and the PI, which is similar to the estimate of Rohling (1991) and is in line with his theoretical approach. In regions of deep water formation in the PI (i.e. the Gulf of Lions, the Adriatic Sea, Fig. A1), the shallowing is more pronounced (> 60 m) due to an increased water column stability. Only the northern Ionian Sea shows a strong deepening of the pycnocline for the LGM, which is linked to the lack of deep water formation in the Adriatic Sea (Fig. A1). During the PI, dense water is formed in the Adriatic Sea, flows across the

Otranto sill, supplies to the deep and intermediate water of the Northern Ionian Sea, and, thus, lifts up the density iso-surfaces (Liu et al., 2021). This deep water formation during the PI leaves its imprint on the water masses of the northern Ionian Sea where colder, less saltier and "younger" water mass properties are found at intermediate water depths in the west of the Ionian basin compared to the east (not shown). At the LGM, water mass properties are very similar across the entire northern Ionian basin. In the sensitivity study PI-Straits, the deep water formation in the Adriatic Sea is slightly reduced, but still active (Fig.





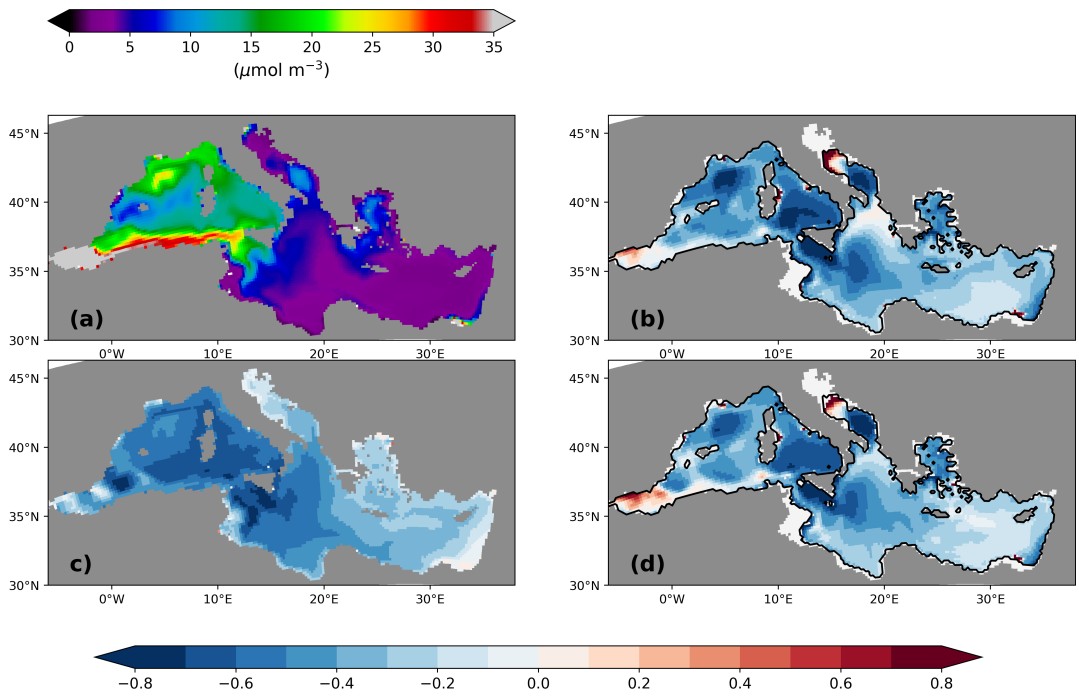

**Figure 11.** Surface phosphate concentration ($\mu$mol P m$^{-3}$) for the PI of GLAC-1D (a, see color scale above panel a) and its relative concentration change between the LGM and the PI for GLAC-1D (b), and for ICE-6G (d). Panel (c) shows the relative concentration change between PI-Straits and the PI of GLAC-1D.

A1e), and, hence, we do not find a corresponding pattern in the pycnocline depth change (Fig. 10). The similarity of the spatial pattern in PI-Straits compared to the LGM simulations underlines that shallower sill depths alone are the largest contributor to the pycnocline anomaly for most areas of the MedSea. The reduced ZOC (Fig. 3) and the consequentially longer water residence time within the basins of the MedSea increases the SSS (Fig. 7) and intensifies the water column stratification in all experiments with LGM sill depth (Fig. 2b, f).

### 4.2 Changes of the biogeochemical distributions in the water column

Following the argument by Rohling (1991), who postulated that a shallower pycnocline provides more nutrients to the production zone and enlarges the NPP, we might expect, according to Fig. 10, an increase in surface phosphate concentration and NPP. In contrast, the simulated surface phosphate concentration at the LGM decreases remarkably by 20 to 60 % compared to its PI value (Fig. 11b, d). Consequentially, the vertical integrated net primary production resembles this pattern change (Fig.

5c, e). Largest changes in surface phosphate and NPP are located in deep water formation areas and the western Ionian Sea. Suppressed or shallower deep water formation hampers the supply of phosphate (PO4) to surface layers. Relative increases of surface phosphate are found only in the Alboran Sea and in vicinity of rivers. The increased river runoff during the LGM (Tab. 2) also transports more nutrients (phosphate, nitrate, silicate) to the MedSea according to our parametrization which is based





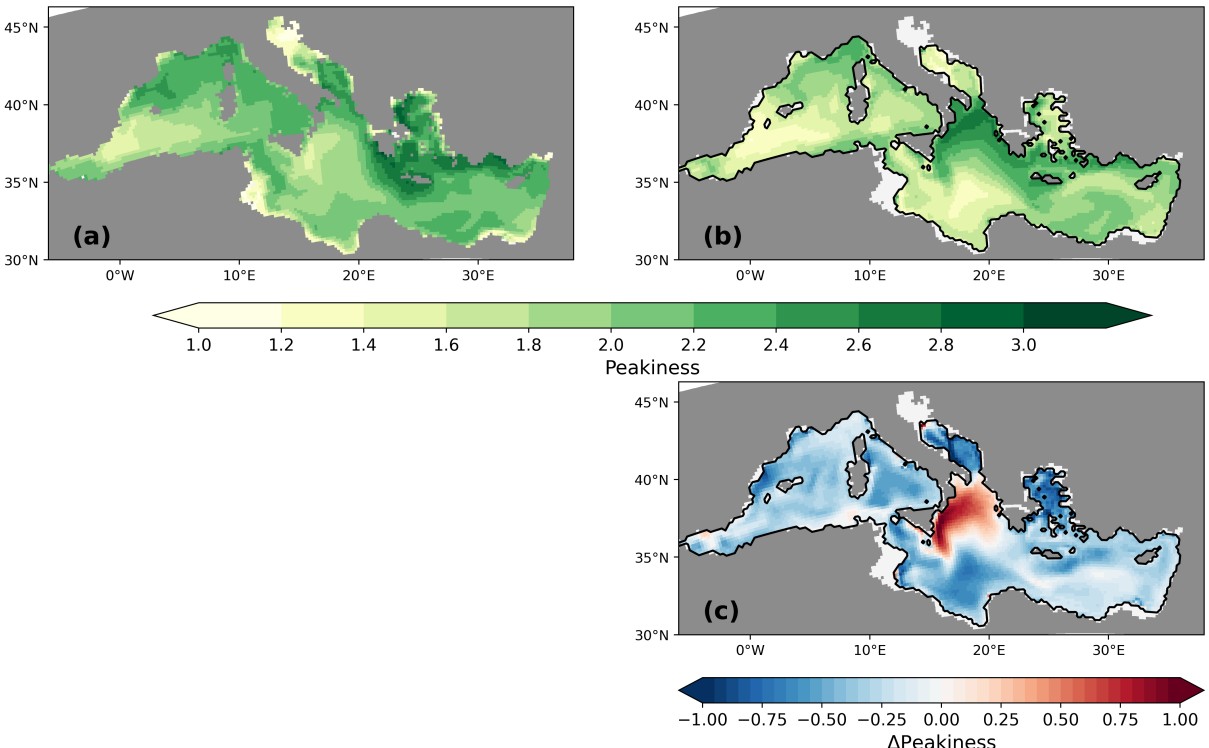

**Figure 12.** Peakiness $P$ for the PI (a) and the LGM (b) of GLAC-1D and the difference between LGM and PI of $P$. See text for definition of peakiness.

on a basin-wide fixed river nutrient concentration times river discharge. The relative increase in NPP in the vicinity of the Nile

river underlines the change in the runoff (Fig. 5c, d). In the northern Adriatic Sea, we see a relocation of the productive area. Bathymetry changes due to the low sea level stand move the coast line and, thus, the river estuaries towards the south. Note, that we find a similar overall relative decrease of surface PO4 for PI-Straits, only missing the extreme changes in deep water formation areas and the northern Adriatic Sea (Fig. 11c).

  Besides an overall decrease to the annual mean NPP, we also find a flattening of the seasonality of the production. To

illustrate the changes of the NPP seasonality, we define the peakiness of the seasonal NPP with

$$P = \frac{12\,NPP_{max}}{\sum NPP_{ann}}$$

where $NPP_{max}$ is the annual maximum of the monthly mean NPP (we use a running mean of two months to ensure that we find a correct maximum within our monthly mean data), and $\sum NPP_{ann}$ is the total annual NPP. In case of the same NPP for each month, $P$ would be exactly 1. For $P=3$, 50 % of the annual NPP is confined to two adjacent months. For the theoretical

case of $P=6$, 100 % of the annual NPP would occur in two adjacent months. The peakiness is, thus, a simple measure to characterize "blooming areas" and "non-blooming areas", independent of the annual mean NPP (see a similar approach by D'Ortenzio and Ribera d'Alcalà (2009)). For the PI, the peakiness is close to 3 in the Aegean and the northern Levantine and well above



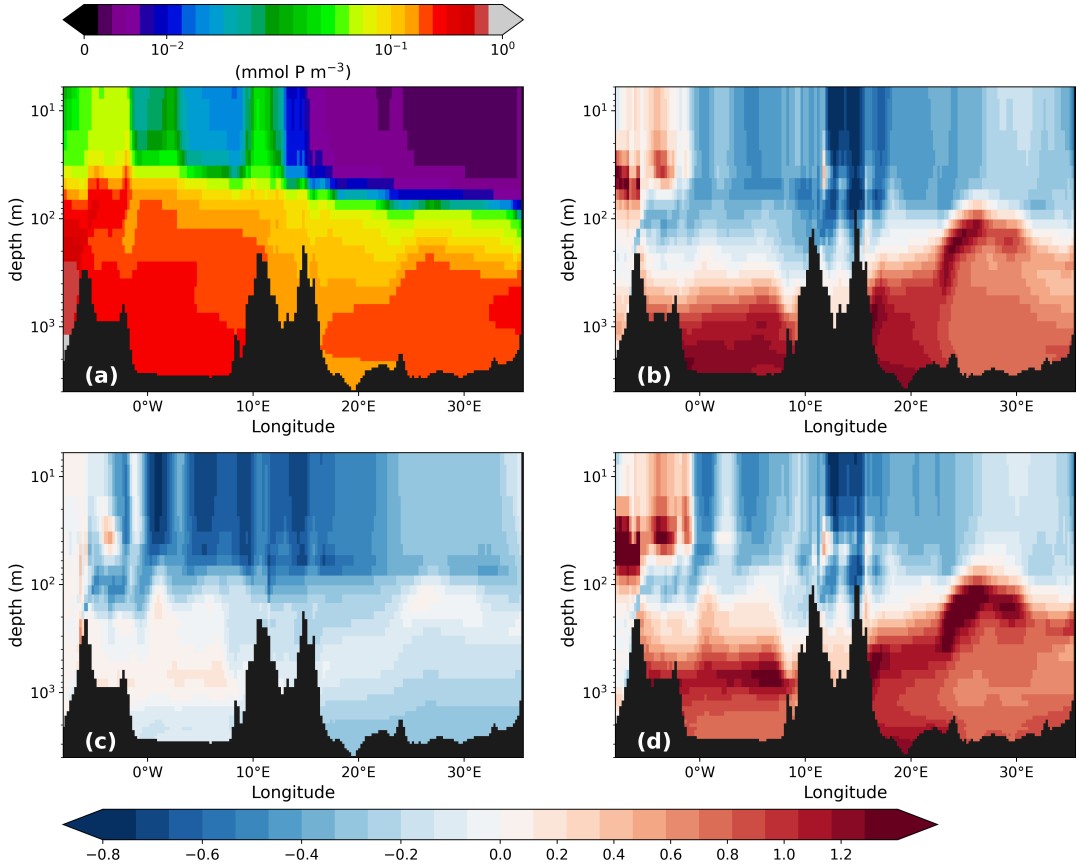

**Figure 13.** Phosphate concentration (mmol P m$^{-3}$) along the east-west transect (for location see Fig. 1) of the PI of GLAC-1D (a) and the relative concentration change between the LGM and the corresponding PI of GLAC-1D (b) and ICE-6G (d), and between PI-Straits and the PI of GLAC-1D (c). Note the logarithmic color scale in panel a) and the logarithmic depth scale in all panels.

2.2 for the Gulf of Lions and large parts of the Levantine indicating a well developed seasonal peak in the production, primarily in spring, and a lower production throughout the rest of the year (Fig. 12). Coastal areas, especially in the vicinity of rivers

show a more uniform production seasonality ($P$ close to 1). This pattern for the PI resembles the findings of D'Ortenzio and Ribera d'Alcalà (2009) who characterized the seasonal cycle of surface biomass for different areas of the MedSea based on satellite data. For the LGM, the seasonality of the NPP flattens throughout the entire MedSea. The northern Ionian Sea is the only exception, where we find a shift towards blooming conditions. The pattern change of the peakiness resembles the changes of the pycnocline (Fig. 10), which might explain the changes in the nutrient supply throughout the year.

To understand the fate of the surface nutrients, we first look at the vertical PO4 distribution. As already visible in the concentration profiles for the GOL and the LEV (Fig. 2), we see a strong accumulation of PO4 below 100 m during the LGM. The east-west cross-section through the MedSea displays sharp vertical gradients around 100 m with relative changes of more than 100 % in both basins and both LGM simulations (Fig. 13). In PI-Straits, an accumulation of PO4 at depth is missing and we





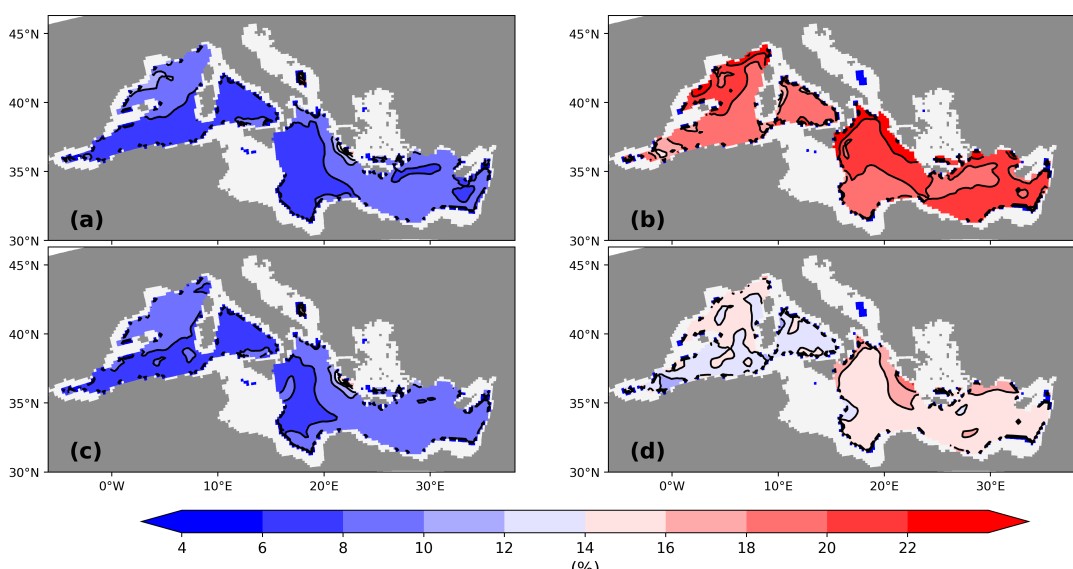

**Figure 14.** Transfer efficiency of detritus calculated as ratio of the detritus flux at 1000 m to 100 m on a percentage basis (%) for the PI of GLAC-1D (a) and for the LGM of GLAC-1D (b) and ICE-6G (d). Panel (c) gives transfer efficiency for PI-Straits.

rather find a nutrient depletion over the entire MedSea. The shallower sill depths in the LGM and PI-Straits lead not only to
a more sluggish overturning circulation (Fig. 4), but also to a shift of the centre of maximum zonal transport to a shallower depth. This causes a very effective lateral export of nutrients and organic material at intermediate depth from the eastern to the western basin and out of the MedSea and, thus, leads to the depletion of nutrients in PI-Straits. In the LGM simulations, the same process is active, but is overcompensated by the effect of an additional change of the remineralization rate of organic material. The remineralisation rate depends primarily on temperature and oxygen availability. Any potential restrictions by oxygen
are negligible in the well ventilated water of the MedSea during the PI and the LGM (Fig. 2). Temperatures at the LGM are significantly lower by -8 to -4 K than in the PI (Fig. 2), causing remineralization rates to drop to 51-64 % of the PI values, respectively. In combination with gravitational sinking, more detritus reach intermediate and deep layers and potentially enter the sediment in the LGM simulations. The transfer efficiency of detritus $T_{eff}$ (Weber et al., 2016) which relates the detritus sinking flux at 1000 m to that of 100 m illustrates this effect of the remineralisation rate (Fig. 14).

In the PI, less than 10 % of the detritus export flux at 100 m reaches the 1000 m depth level. This is also true for PI-Straits which has a nearly identical vertical temperature distribution as the corresponding PI simulation (differences of less than ±0.5 K between PI-Straits and PI of GLAC-1D, not shown). In contrast, $T_{eff}$ increases to 14-28 % in both LGM simulations. Higher $T_{eff}$ in the LGM of GLAC-1D than in ICE-6G is attributable to the overall colder conditions in this simulation (Fig. 2, Fig. 7). To further illustrate the combined effects of shallower sill depths and colder temperatures on the nutrient distribution, we
present, as an example, the dynamical conditions in the Strait of Sicily for GLAC-1D (Fig. 15). The mean transport of total phosphate (i.e. the sum of dissolved and organically bound phosphate) across the strait is southward in the upper 137 m for the



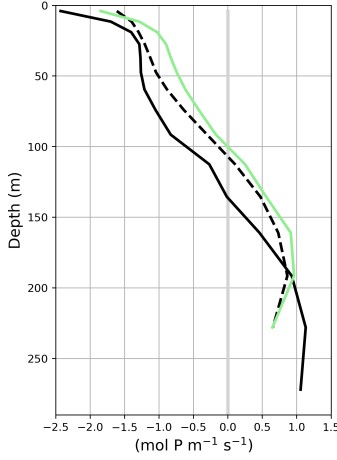

**Figure 15.** Vertical depth profile of the meridional transport of the sum of dissolved and organically bound phosphate (mol P m$^{-1}$ s$^{-1}$). over the Strait of Sicily for the PI (solid black line), the LGM (dashed black line) of GLAC-1D and PI-Straits (solid green line). Positive values indicate northward transport, i.e. from the eastern into the western basin of the MedSea.

PI, with a corresponding northward return flow below. With the shallower strait depths for GLAC-1D-LGM and PI-Straits, the interbasin exchange fluxes are, as expected, reduced and the neutral position between the baroclinic flow components shifts to 111 m and 97 m, respectively. Below this depth, both simulations show a more effective export of total phosphate from the

eastern basin as compared to GLAC-1D-PI. In combination with a reduced inflow of total phosphate into the eastern basin, this leads to the simulated nutrient depletion in the upper 200 m of the water column as seen in Fig. 13. The warmer temperatures in PI-Straits than in the GLAC-1D-LGM allow for a higher biological production despite similar near-surface phosphate concentration, as indicated by slightly higher organically bound phosphate concentration in the upper ocean (Fig. A4b). However, the temperature effect on remineralization manifests in a more rapid decay of detritus and a higher phosphate concentration

between 100-200 m in PI-Straits compared to the LGM simulation of GLAC-1D (Fig. A4a), eventually leading to the even larger nutrient depletion of the upper ocean.

In brief, the impact of shallower sill depths is primarily a shallowing of the overturning circulation which leads to a more effective zonal export of nutrients from intermediate ocean depths. As seen for PI-Straits, the phosphate inventory of the entire MedSea consequently decreases even under present day climate conditions. Furthermore, a potential increase in net primary

production due to a shallower pycnocline, as proposed by Rohling (1991), is outweighed by this efficient drainage of nutrient from the MedSea. For the colder LGM conditions, we find a comparable decrease in surface PO4 concentrations (Fig. 5) and NPP (Fig. 13), but temperature-induced lowering of remineralization rates leads to a higher $T_{eff}$ and a phosphate accumulation at greater depths. It is important to emphasise that a higher $T_{eff}$ during the LGM masks the signal of the lower surface net primary production and leaves its imprint in the sediment. The implications will be discussed in the next section.






### 4.3 Changes in the sediment composition for the LGM

As mentioned, changes in the remineralisation rate also modify the deposition flux to the sediment. Relative changes of the settling flux rate, especially in the deep Ionian Sea, are 5-10 times higher than during the PI, while NPP decreases to -0.4 to -0.6 in the same area (Fig. 5). Fig. 6 displays the changes in weight-percentage sediment composition between the LGM and the PI. Both LGM simulations show an increased fraction of detritus and opal in the upper 6 mm of the sediment column (first two model layers of the simulated sediment column) with at least a doubling of the local contribution of both components almost over the entire MedSea. Changes in the deep basins are more pronounced due to the higher transfer efficiency at the LGM (factor of ten). Again, the even higher transfer efficiency due to colder temperatures of GLADC-1D is evident from the larger fractional change in detritus and opal. The calcite fraction correspondingly decreases throughout the MedSea with largest changes in the Ionian Sea and the Adriatic Sea (< 0.5) and more moderate changes in the Alboran Sea (Fig. A2). Sediment core data confirm 20-50 % lower than modern carbonate contents during the LGM for different areas of the Mediterranean Sea (Aksu et al., 1995; Hoogakker et al., 2004). Please note, that in our simulation a large part of the weight-percentage change in calcite is related to the higher dust deposition during the LGM, especially in the northern part of the MedSea (Fig A2, Albani et al. (2016)).

Higher accumulation of organic carbon during the LGM as compared to the PI is supported by an increased abundance of benthic foraminiferal tests which develops under elevated food supply of detritus (Schmiedl et al. (2010) and refs. within). Hence, our simulated sediment changes are in good agreement with sediment proxies. Moreover, it has to be emphasized that these elevated sedimentary fluxes occur under a simulated lower NPP. Thus, our results link the supposedly contradictory statements of a higher NPP during the LGM as inferred from benthic proxy data (Schmiedl et al., 2010) and a lower NPP as derived from the absence of small placoliths, a coccolithophoridae used as classical indicator for high nutrient availability (Ausín et al., 2015).

### 4.4 Assessment of the biological production temperature T$_{org}$

The use of the alkenone unsaturation ratio U$_{37}^{K'}$ to estimate the annual mean sea surface temperature is a widely applied paleo-ceanographic tool (Müller et al., 1998; Conte et al., 2006). Potential seasonal biases which arise from the marked seasonality in the abundance of primary producers and the link between primary production and surface ocean temperatures, have been a debatable issue, but, for most of the ocean regions, Rosell-Melé and Prahl (2013) concluded that "the integrated sedimentation patterns for U$_{37}^{K'}$ measured in sediment trap time series provide a measure of annual mean SST" and that a seasonal pattern is not apparent in a temperature reconstruction with U$_{37}^{K'}$ in a global application. However, all derived relationships for U$_{37}^{K'}$ are based on modern conditions of spatial and temporal surface ocean temperature pattern, as well as, on present day variations in biological production. So far, there are no data or methods to assess these relationships for past climate conditions.

Our diagnostic tracer T$_{org}$ (see Sec.2) mimics the information which is obtainable from alkenones without the method-ological and physiological uncertainties typical of alkenone undersaturation estimates (Conte et al., 2006). Thus, T$_{org}$ gives us the opportunity to analyse, i) how well it captures the surface ocean temperature change between past and present, and ii)





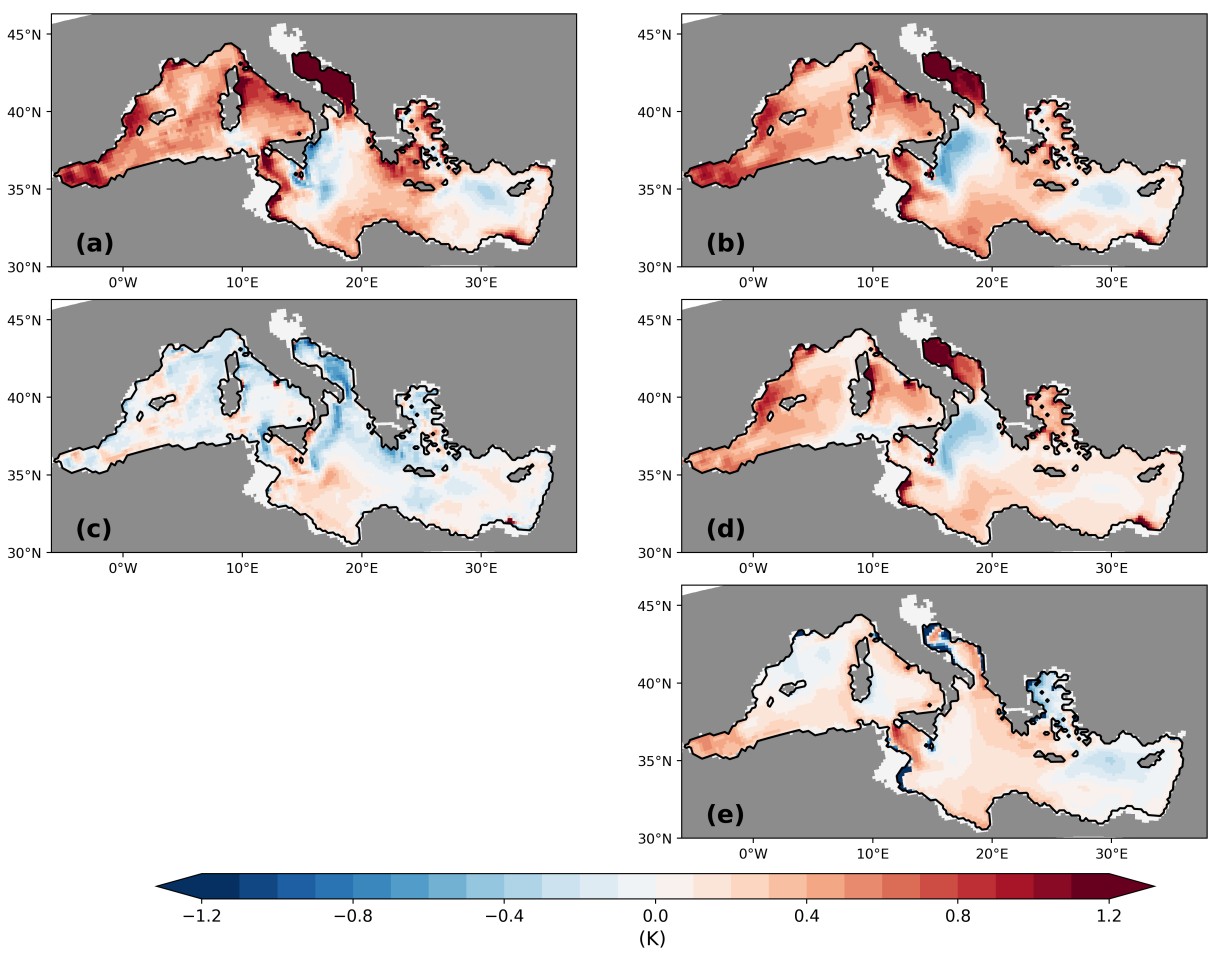

**Figure 16.** Difference of $\Delta T_{org}$ and $\Delta T_{32}$ (a). Note, that both anomalies are negative; thus, positive values in (a) indicate a larger absolute $\Delta T_{32}$. Also shown is the difference of $\Delta PWT$ and $\Delta T_{32}$ (b) and the difference between $\Delta PWT$ and $\Delta T_{org}$ (c). Panel (d) shows $\Delta PWT_{NPP}$, the NPP related contribution to $\Delta PWT$, and panel (e) displays $\Delta PWT_T - \Delta T_{32}$, the temperature related contribution. All units are in (K). See text for definitions.





which processes might contribute to the difference of the signals between $T_{org}$ and temperature. To tackle these questions, we
calculate the change in $T_{org}$ between LGM and PI of GLAC-1D ($\Delta T_{org}$) and the corresponding change in the surface ocean
temperature, for which we use the annual mean temperature of the upper 32 m ($\Delta T_{32}$, i.e. corresponding to the first 4 layers of
our model), where most of the simulated biological production takes place.

Fig. 16a shows the difference of $\Delta T_{org}$ and $\Delta T_{32}$. Note, that we here discuss only the analysis for GLAC-1D. The corre-
sponding figure for ICE-6G is given in the appendix for the sake of completeness (Fig. A5). For most parts of the MedSea,
the cooling signal that is captured by $\Delta T_{org}$ is smaller than $\Delta T_{32}$, resulting in an absolute positive difference because both
anomalies are negative. In the western MedSea, the absolute value of $\Delta T_{org}$ underestimates that of $\Delta T_{32}$ by 0.4-1 K, and even
higher differences are found in a selected areas, e.g. the Adriatic Sea and close to the shelf regions in the western Ionian Sea.
Our result indicates that $T_{org}$ does not just simply reflect $T_{32}$, but that additional processes must shape $T_{org}$. This might be the
seasonality of the net primary production or processes within the water column such as the remineralization of organic matter
during settling. The latter was postulated by Conte et al. (2006) who showed that different remineralization functions alone led
to changes of $T_{org}$ between 0.4-0.9 K in their hypothetical case studies.

To gain insight into the drivers of $T_{org}$, we construct a diagnostic production-weighted temperature (PWT) by integrating
the product of the monthly mean NPP and the monthly mean temperature over all layers and all months and dividing the
result by the vertical integrated annual NPP. The difference between PWT for the LGM and the PI ($\Delta PWT$) reflects changes
in the NPP and its temperature recording without any impact from remineralization or settling processes. We find a strong
similarity between the patterns of the difference of ($\Delta PWT$ -$\Delta T_{32}$) and ($\Delta T_{org}$-$\Delta T_{32}$), which underlines that changes in the
surface conditions are the major drivers of $T_{org}$ variations (Fig. 16a, b). This becomes even more obvious when we separate the
variations of $\Delta PWT$ into a part attributed to variations in the NPP ($\Delta PWT_{NPP}$) and a part attributed to temperature variations
($\Delta PWT_{T}$). For $\Delta PWT_{NPP}$, we calculate the difference of the present day temperature times the monthly mean NPP of the
LGM and the present day temperature times the monthly mean NPP of the PI. Thus, we get variations based only on the shift
of the NPP seasonality, which results in a changed sampling of the monthly mean temperatures. For calculating $\Delta PWT_{T}$, we
use the present day NPP times LGM temperatures and present day NPP times PI temperatures. This gives the contribution
from the fact that the annual mean LGM cooling signal is not distributed uniformly over the year. In our GLAC-1D simulation,
the Levantine shows a stronger cooling signal than the annual average during February to April, while in the Ionian Sea, the
temperatures of July to September primarily contribute to the annual mean LGM cooling signal (not shown).

Fig. 16d shows that the pattern of $\Delta PWT_{NPP}$ explains most of the signal found in $\Delta PWT$, including the cooling signal in
the northern Ionian Sea. Furthermore, $\Delta PWT_{NPP}$ resembles the pattern of the peakiness change (Fig. 12) with a warming
signal in regions with decreasing peakiness and vice versa. This clearly indicates that the NPP driven part is the major driver
of PWT changes. The temperatures related part $\Delta PWT_{T}$ has a minor contribution to $\Delta PWT$ (Fig. 16e), but it explains e.g. the
small cooling signal west of Cyprus. As mentioned before, the absolute spring temperature change in the Levantine is stronger
than the annual mean LGM cooling signal. Other potential sampling biases, such as vertical shifts in the production or due to
sampling over shallower water depth in the LGM, have no impact on $\Delta PWT$ (not shown). The contribution from processes
within the water column, e.g remineralization or lateral advection, are estimated by the difference between $\Delta PWT$ and $\Delta T_{org}$





(Fig. 16c). For most of the MedSea, this contribution is rather small, except for regions where we also find strong changes in

the physical ocean, such as the northern Ionian Sea, the Adriatic Sea, or the Strait of Sicily. A similar pattern for $\Delta$PWT-$\Delta$T$_{org}$ is seen for ICE-6G (Fig. A5c). We can not disentangle whether the cause of this signal in T$_{org}$ is related to the lateral advection of organic material from distant production regions or a temperature-induced remineralization change or the combination to both.

To summarize, we find that the recorded temperature of $\Delta$T$_{org}$ underestimates the climate signal which is represented by

455 $\Delta$T$_{32}$. The shift in the seasonality of the NPP between past and present is a main driver of the difference between $\Delta$T$_{org}$ and $\Delta$T$_{32}$, as a more uniform seasonal cycle in NPP tends to sample the relative warmer temperatures of late spring and summer. Despite the simplicity of our artificial temperature tracer T$_{org}$, our simulations might point to the methodological problem of using present day derived correlations for paleo-records. Furthermore, local changes in the biogeochemical patterns increase the uncertainty that is intrinsic to the method (Conte et al., 2006).

## 5   Summary and Discussion

We apply a regional physical-biogeochemical ocean model of the Mediterranean Sea to investigate the biogeochemical state during the LGM. The use of a novel set of atmospheric and oceanic boundary conditions derived from paleo-simulations over 26,000 years with an Earth system model (MPIESM, Kapsch et al. (2022)) allows for running consistent time slice simulations for the LGM and the PI. To address uncertainties for the LGM, we take forcing data from two MPIESM runs based on different

ice sheet reconstructions (Tarasov et al., 2012; Peltier et al., 2015). The computational efficiency of our calculations enables transient model simulations over 2000 years, which minimizes model drift and permits to attribute the simulated changes to physical and biogeochemical variations during the LGM. Our analysis is supported by a sensitivity study with present day climate and LGM strait bathymetry (PI-Straits) to disentangle topography driven and climate induced changes on the biogeochemistry.

Simulated circulation changes for the LGM are primarily caused by sea level low stands which lead to a weakening of the zonal overturning circulation and an intensification of water column stability. Our model results for the PI and the LGM are in line with previous ocean modelling attempts (Myers et al., 1998; Mikolajewicz, 2011; Sevault et al., 2014; Grimm et al., 2015) and theoretical considerations (Rohling et al., 2015). Simulated changes in sea surface temperature fall into the range of estimated surface cooling (Hayes et al., 2005).

The similarly strong response of the biogeochemistry in both LGM simulations indicates the robustness of the signals such as the nutrient accumulation below 100 m in the western and eastern basin of the MedSea. A quantitative evaluation of our simulated LGM distributions is still challenging. The evaluation of our results on the changes in the biogeochemistry during the LGM is restricted to the comparison to sediment core data, as to our knowledge there is no other published modelling study for the MedSea on this topic. As deduced from the sediment archives, an observed increase in diversity and the proportion

of infaunal benthic foraminifera supports a glacial increase of organic matter fluxes to the sea floor (Schmiedl et al., 1998; Kuhnt et al., 2007; Abu-Zied et al., 2008; Schmiedl et al., 2010). Our model simulations confirm a higher detritus deposition,





but this emerges from lower surface net primary production. Nutrient availability at the surface is reduced due to higher water column stability, a more efficient lateral nutrient export at mid-depth out of the MedSea, and a larger transfer efficiency of detritus at 1000 m under the cold LGM climate. A simulated lower production during the LGM is in agreement with data of sediment composition which show an absence of small placoliths, a coccolithophoridae used as classical indicator for high nutrient availability (Ausín et al., 2015). The occurrence of higher depositional fluxes despite lower primary production may clarify supposed inconsistencies in the interpretation of sediment data.

Our results emphasize the non-linearity of the response of the biogeochemistry to the hydrodynamical changes at low sea level stand. The assumption that the shallowing of the pycnocline inevitably introduces higher net primary production (Rohling, 1991) must be revised, and the interplay of altered lateral transport and reduced biological turnover rates must be considered for the LGM. This is underlined by the findings from the sensitivity experiment PI-Straits. By only reducing the water inflow from the Atlantic due to a shallower sill depth, most of the biogeochemical pattern change in the upper ocean, which is found for the LGM, is reproduced under present day climate. Our findings clearly point to the need of long-term modelling attempts, ideally with a suite of different ocean biogeochemistry models, to gain further insights into the mean state and the temporal variations of the last glacial.

A noteworthy additional outcome of our simulations is provided by the artificial temperature tracer $T_{org}$. Changes in temperature of the upper 32 m between LGM and PI are imprinted on the sediment with a spatial heterogeneous pattern. In the western Mediterranean Sea, the $T_{org}$ signal underestimates the LGM upper ocean temperature anomaly by about 1 K, while $T_{org}$ records a stronger cooling of up to 0.5 K for the northern Ionian Sea. Thus, even within our ideal model world where all physiological and methodological uncertainties of temperature recording by planktic organisms (Conte et al., 2006) are omitted, the $T_{org}$ signal captures the upper ocean temperature with a large uncertainty. Please note, that we do not claim that we correctly represent alkenone production, but we simply track the temperature at phytoplankton growth and the fate of the signal carried by organic material on its way to the sediment.

The newly developed model framework is here used to understand the biogeochemical state at the LGM. The resulting distributions will serve as starting points for simulations covering the entire last deglaciation to present, with a main focus on the formation of sapropels in the eastern Mediterranean Sea during the early Holocene. As has been shown by time slice simulations (Grimm et al., 2015), the occurrence of sapropels does not depend only on "short-term" variations of biogeochemical and hydrodynamical conditions, but requires a long prelude of deep-water stagnation. Our new model framework in combination with the consistent forcing data sets from the MPIESM (Kapsch et al., 2022) provide a good working tool to tackle this topic.

*Code and data availability.* Model data and Python scripts used for the analysis and visualization are available on Zenodo (Six, 2024). Proxy data must be requested directly from Hayes et al. (2005) and Lee (2004). Model code of medHAMOCC is available upon request from the corresponding author.



# Appendix A: Additional Figures for GLAC-1D, ICE-6G, and PI-straits

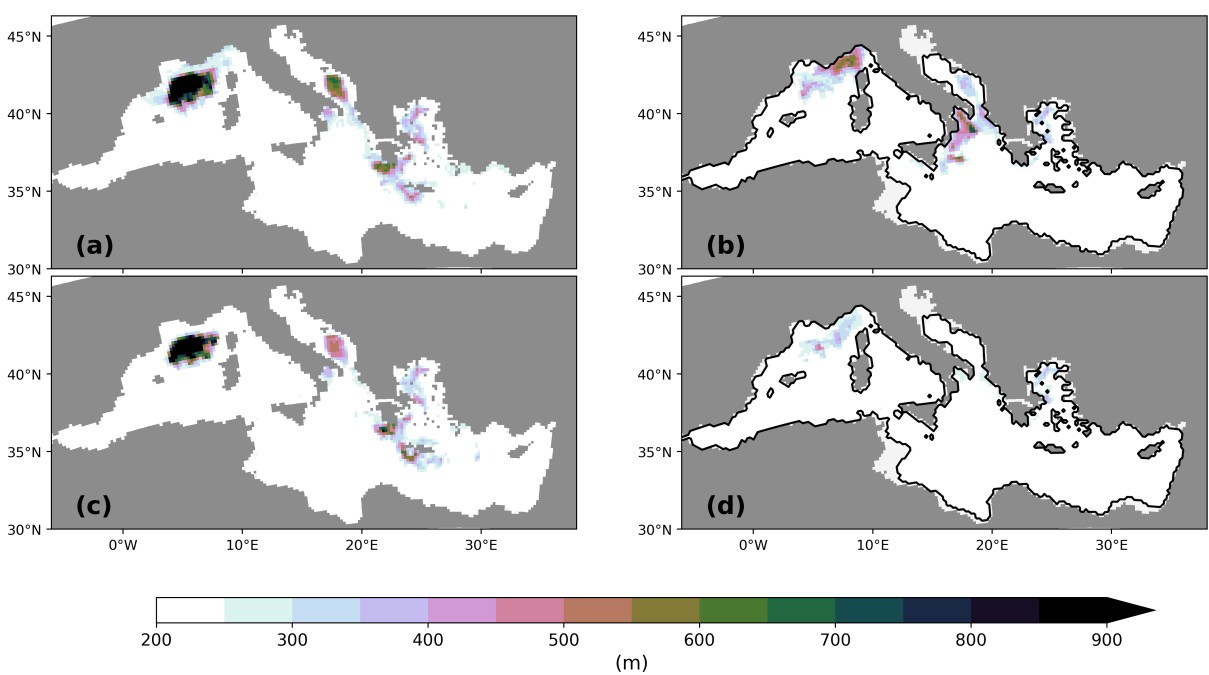

**Figure A1.** Maximum annual mixed layer depth (m) for the PI of GLAC1D (a) and for the LGM of GLAC-1D (b) and ICE-6G (d). Panel (c) give the information for PI-Straits. The contour line indicates the LGM coastline.



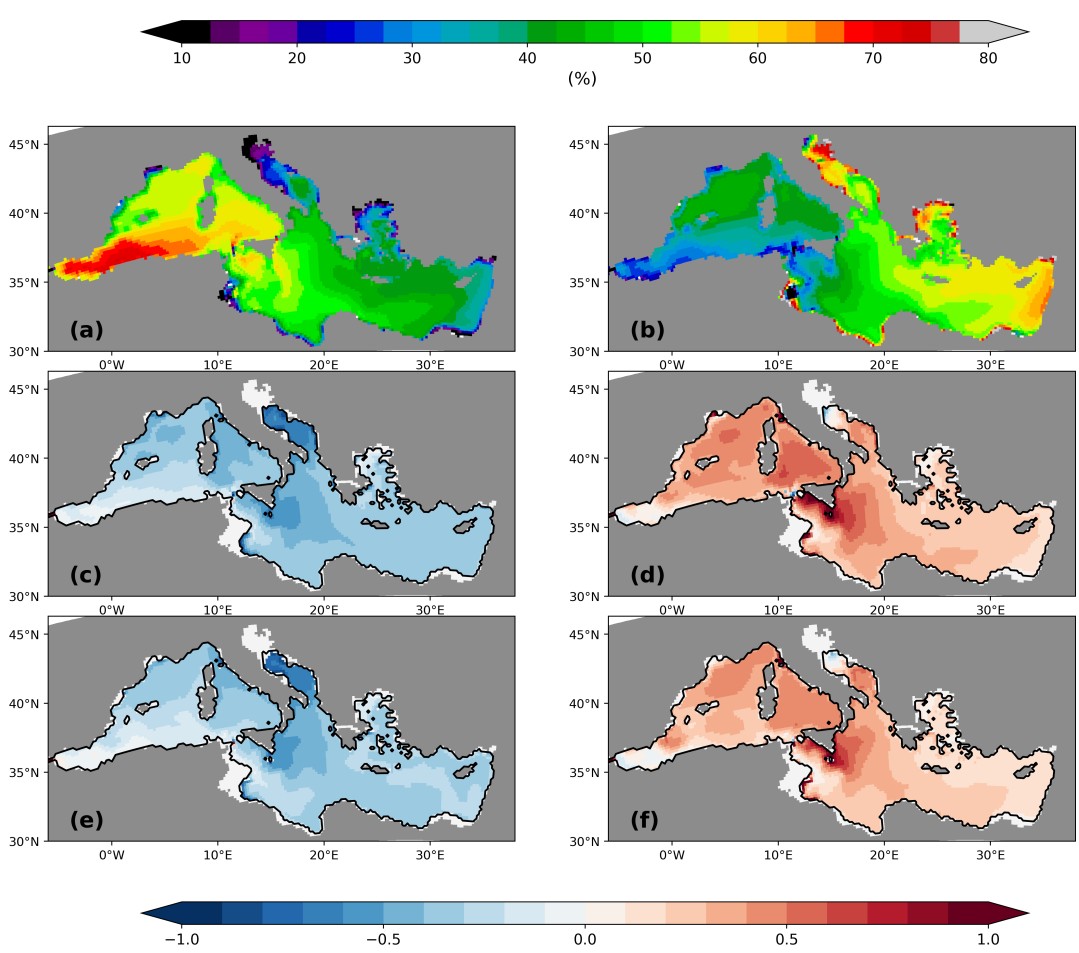

**Figure A2.** Weight-percentage sediment contribution in the upper 6 mm of the sediment column of calcite (a) and terrestrial material, i.e. clay, (b) for the PI of GLAC-1D (in %). Relative changes between LGM and the corresponding PI are given for calcite (c, e) and clay (d, f) for GLAC-1D (c, d) and ICE-6G (e, f).





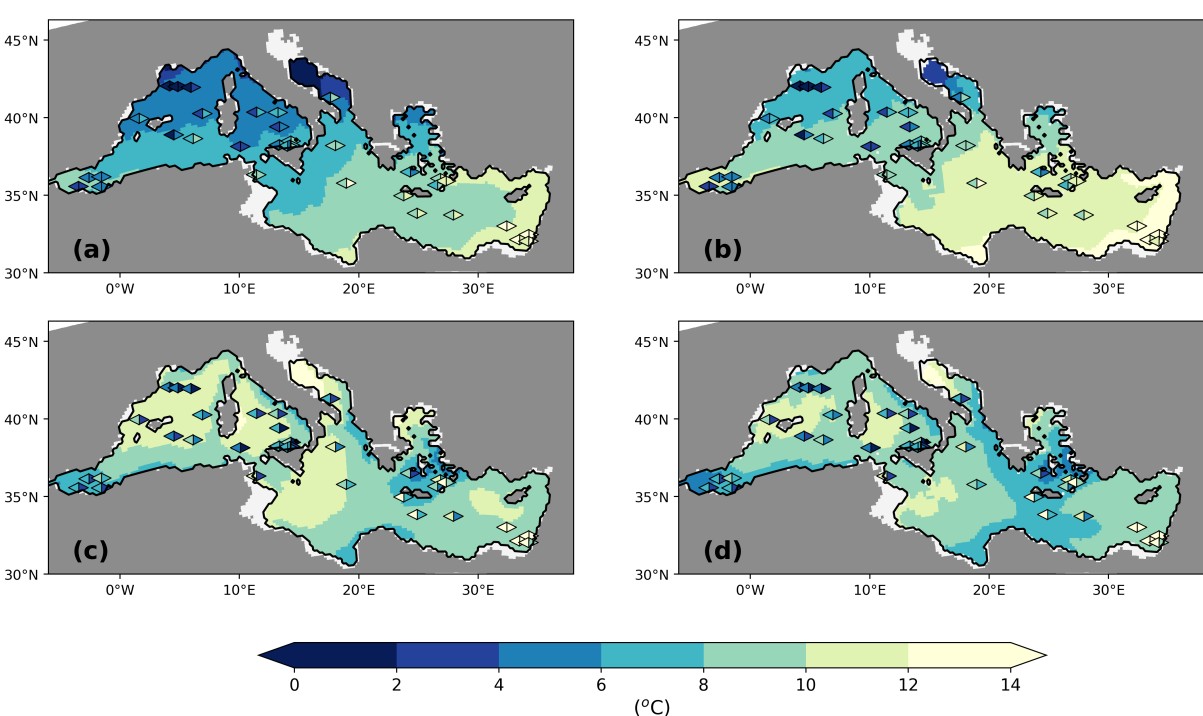

**Figure A3.** Annual minimum sea surface temperature (SST, a, b) and seasonal amplitude of SST (c, d) at the LGM for GLAC-1D (a,c) and ICE-6G (cbd). Seasonal amplitude is calculated from annual maximum and annual minimum. All units are in °C. Overlaid data from Hayes et al. (2005) for the same quantities estimated with two different methods: artificial neural network(left part of rhombu s) and the revised analogue method (right part of rhombus). See Hayes et al. (2005) for more details.

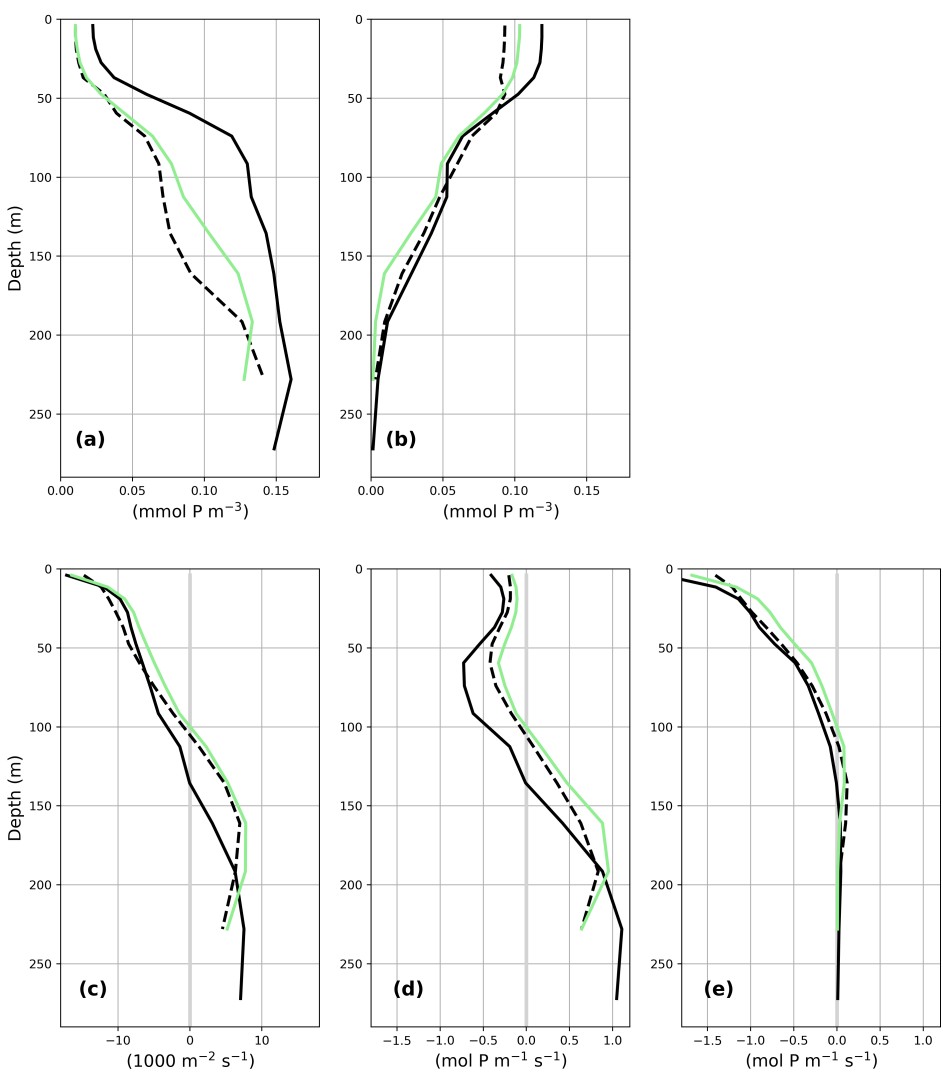

**Figure A4.** Depth profiles for the Strait of Sicily of the concentration of dissolved (a) and organically bound phosphate (b), both in mmol $m^{-3}$, the meridional transport through the strait (c, velocity times straits width, in $1000\,m^2\,s^{-1}$), and the transport for the individual phosphate contributions which are combined shown in Fig. 15, as there is transport of dissolved phosphate (d), and transport of of organically bound phosphate (e), both in mol $m^{-1}\,s^{-1}$. Color coding as Fig. 15: GLAC-1D-PI (black solid line), GLAC-1D-LGM (black dashed line), and PI-Straits (green line).

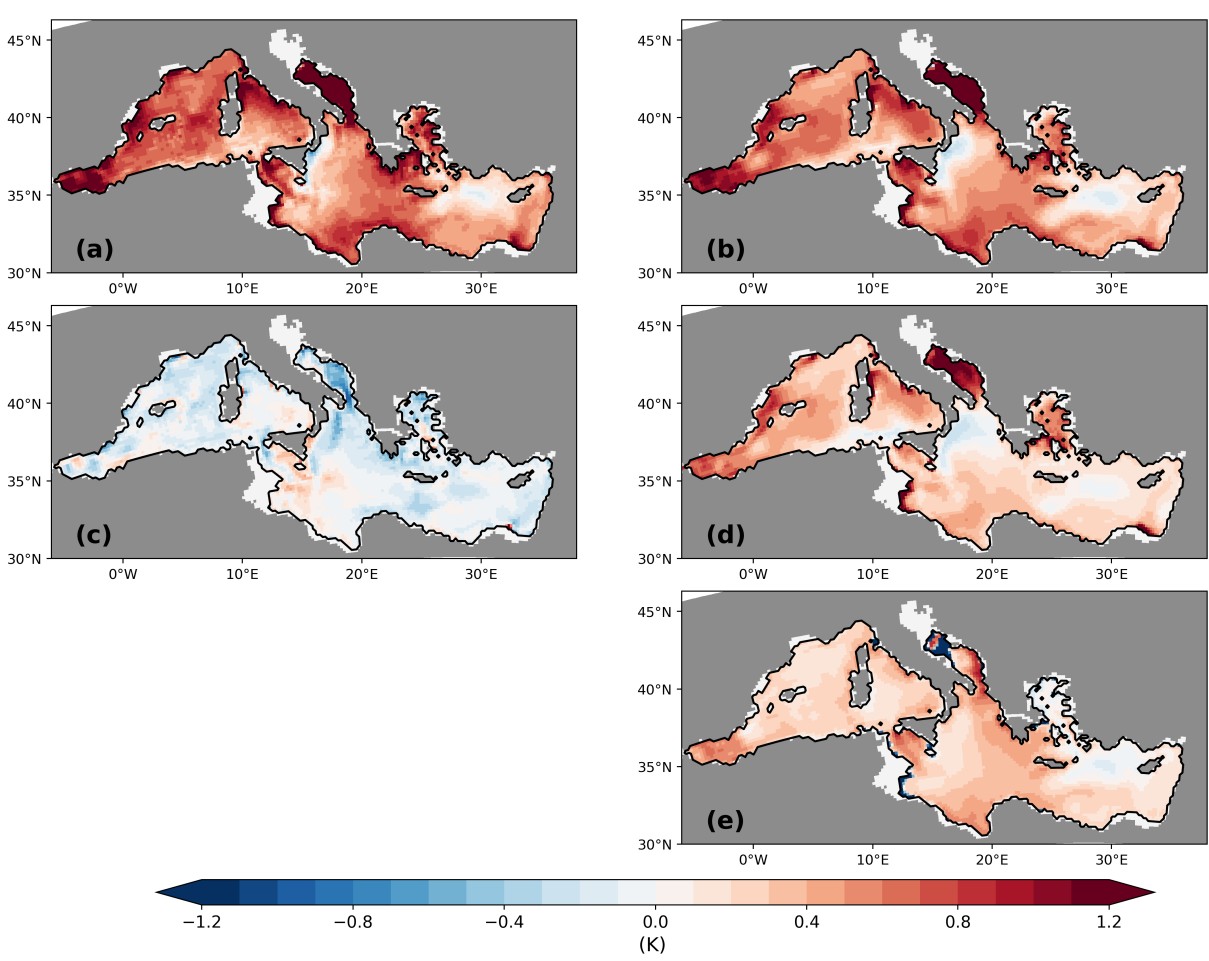

**Figure A5.** Same as Fig. 16, but for ICE-6G. Shown are the difference of $\Delta T_{org}$ and $\Delta T_{32}$ (a), the difference of $\Delta PWT$ and $\Delta T_{32}$ (b) and the difference between $\Delta PWT$ and $\Delta T_{org}$ (c). Panel (d) shows $\Delta PWT_{NPP}$, the NPP related contribution to $\Delta PWT$, and panel (e) displays $\Delta PWT_T$ - $\Delta T_{32}$, the temperature related contribution. As for GLAC-1D, $\Delta T_{org}$ does not capture the full cooling signal given by $\Delta T_{32}$.



*Author contributions.* U.M. and K.D.S. designed the experimental setups. K.D.S. performed and analysed all simulations. All authors criti-
515 cally discussed the presented results and contributed by providing valuable feedback during the manuscript compilation.

*Competing interests.* The authors declare that they have no conflict of interest.

*Acknowledgements.* All simulations were performed at the German Climate Computing Center (DKRZ). Katharina D. Six is funded by the
Deutsche Forschungsgemeinschaft (DFG, German Research Foundation) under Germany's Excellence Strategy – EXC 2037 'CLICCS -
Climate, Climatic Change, and Society' – Project Number: 390683824.



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
