# Peer review of "Modeling Mediterranean ocean biogeochemistry of the Last Glacial Maximum"

_Climate of the Past, 2024_

## Author Comment (AC1)

**Reply to Reviewer #2**

We thank reviewer #2 for the detailed comments in a very friendly manner. We addressed all points with our answer given in Italics following the original comment.

The authors reported original results using a regional physical-biogeochemical ocean model to study the Mediterranean Sea circulation and biogeochemical cycle during the LGM. The regional model was forced by the data of Earth System model taking into account topography and bathymetry changes, which is highly appreciated. Several interesting findings were shown, in particular the sensitivity test attesting the importance of the sill depth in the Straits of Gibraltar and Sicily that explained a large part of sluggish Mediterranean circulation at the LGM. The authors succeeded in reconciling apparent contradiction of proxy reconstruction showing reduced primary production in the surface and increased organic deposition on seafloor during the LGM by the interplay of water column stratification, zonal advection and slower respiration in glacial cold waters. Also, the potential influence of seasonality changes during the LGM on paleo-temperature reconstruction was discussed.

I enjoyed reading this manuscript. The scientific theme fits perfectly with the topics treated by Climate of the Past and this is the first attempt of simulation of biogeochemical cycle in the Mediterranean Sea under glacial conditions. I wish to see this work published on Climate of the Past.

The manuscript is well structured with clear figures and tables. I am not a modeller and I have only limited comments / suggestions.

My first point is scarce comparison between proxy reconstruction and simulation results. Only SST reconstruction based on planktonic foraminiferal assemblages (Hayes et al., 2005) was used for comparison (Figs 8 and 9). The authors mentioned possible bias of reconstruction due to seasonality changes at the LGM taking alkenone as an example but the comparison between model results and UK'37-SST is not presented except for Fig. 9. I would suggest to add a map presenting the comparison using the data provided by Cacho et al. (2002) and Essallami et al. (2007) by updating the MARGO database used in the present manuscript. It will be interesting to examine whether the sites with large SST offset between UK'37-SST and delta T32 correspond to the areas of marked delta peakiness (Fig. 12).

*We updated the data set on UK'37-SST shown in Fig. 9 which is primarily used to evaluate our model results against proxy estimates. However, we refrain from a further assessment between the simulated biological production temperature (T_org) and UK'37-SST. Alkenones are produced by haptophyte algae which have a distinct seasonality affecting the recorded signal (Tierney & Tingley, 2018 ,doi.org/10.1002/2017PA003201). Our simulated primary production does not distinguish between plankton species. Thus, the simulated T_org signal is different to that from proxy data. The focus of our study on T_org is to investigate potential signal differences between T_org and simulated SST which occur in an ideal model framework.*

The second point is about remineralization rate of organic matter and dissolved oxygen concentration. Slower remineralization rate in glacial cold water is proposed to increase transfer efficiency of detritus (Teff). Together with cold sea surface that increase oxygen solubility, generally high dissolved oxygen is expected but some cancelation due to sluggish

ventilation is possible. Therefore, the distribution of dissolved oxygen concentration is highly interesting but only vertical profiles from the selected areas are shown in Fig. 2. I would like to see transect of dissolved oxygen under different simulation scenarios at least in supplementary information.

*For completeness, we included the transect of oxygen in the Appendix Fig A4.*

I would suggest to accept this manuscript after minor revision.

Minor/specific comments

Line 28 and throughout the text. "Gibraltar" would be replaced by "the Strait of Gibraltar".

*done*

Line 28. "ZOC". Do you mean "zonal overturning circulation"?

*done*

Line 33. "Proxy data from foraminiferal shells" should be replaced by "Water mass proxy data recorded in foraminiferal authigenic fraction".

*done*

Lines 155-156, about the nutrient supply from shelf areas during glacial sea level low stands. Was the similar effect considered inside of the Med Sea?

*We refer here to a sensitivity study by Palastanga etal, 2013 which addressed the impact of additional nutrient supply from continental shelf areas in a global setup. According to their paper, they applied the changed nutrient supply globally, which should include the Med Sea. However, the resolution of their global model is very coarse, i.e. it does not have any ocean points less than 200m which could be considered as shelf areas, and the Mediterranean Sea is only represented by a couple of grid cells – I doubt that they looked at regional results at all. In our setup, potential changes in water-sediment interactions due to sea-level low stands within the Med Sea are considered.*

Line 160. About the use of the present-day nutrient concentration for the LGM simulation. I understand the reasoning of the authors but some sensitivity tests by modifying nutrient concentration will be interesting to examine the robustness of the results.

*We performed one extreme sensitivity study by prescribing a tenfold nutrient concentration of PO4 and NO3 compared to present day at the western boundary for the LGM simulation of GLAC-1D. Comparing these results with the LGM simulation with PI values at the western boundary, named here LGM-CTL, we see a significant accumulation of nutrient in the upper 200m (higher than 100 times LGM-CTL in the eastern basin). However, net primary production is higher by only up to 12 times LGM-CTL which is a result of a significant higher standing stock of zooplankton (>30 times) that controls phytoplankton concentrations. Below 200m, we find an almost linear response of the nutrients (~ 10 times LGM-CTL values which corresponds to the prescribed perturbation). Detritus also shows 10 times higher LGM-CTL*

*values down to ~400 m. Below that depth, detritus starts to accumulate because remineralization is slowed down due to the lack of oxygen (less than 5% of the LGM-CTL). At the end of the 1000 years of simulation, the deep basins nearly turned anoxic. It is interesting to find that this extreme, highly unrealistic sensitivity study shows a nearly linear response of the nutrient concentrations to the western boundary at the depth of the LIW and below in the eastern basin. However, we don't see a possibility to provide limits for the open boundary concentrations, besides that ten times higher values produce deep sea anoxic and are therefore very unlikely.*

Lines 185-186, "PEM = precipitation + river runoff - evaporation". The first letter of the words does not correspond to the abbreviation.

*We change it to PR-E = precipitation + river runoff - evaporation*

Line 187 and throughout the text, "1000 m3 s-1". The authors may use always "Sv" or "mSv" because small fluxes like +0.06-0.076 Sv and +0.04-0.11Sv (line 194) are shown with Sv unit.

*We follow the convention to use 1000 $m^3s^{-1}$ for freshwater fluxes in atmospheric sciences and Sverdrup (Sv) for oceanic fluxes, as is common in oceanography. We introduce Sv when it is used the first time.*

Lines 206-207, "slightly weaker...other modelling studies". Add the range of flux obtained by other studies to be more precise.

*We provide a range (0.1-0.3Sv) which is taken from the four mentioned references.*

Line 212, "Nile damming". Do you mean "the Aswan High Dam in 1964"? Please revise.

*Revised.*

Lines 233-234, "Hamann et al., 2008" is missing in the list of reference. The authors may use the map provided by Venkatarathnam and Ryan (1971) that presents a detailed distribution of calcium carbonate in the eastern basin sediments.

*Thanks for pointing out the missing ref. of Hamann etal 2008. We also included the reference to Venkatarathnam and Ryan (1971).*

Line 247, "Kuhlemann et al., 2008". The data provided by this reference is not used for the data model-comparison. Please revise.

*It is correct that the data model-comparison which is shown in Fig. 8 is only based on Hayes et al. (2005). We still keep the reference of Kuhlemann et al 2008 to cite a second independent estimate of the overall SST change in the LGM.*

Line 253. The authors may cite Fig. 8 in addition to Fig. 9.

*Done*

Line 260, "Fig. 7d". Isn't it "Fig. 7b"?

*Correct. Now changed to 7b.*

Lines 304-305. Fig. A1e does not exist. Please revise.

*Correct. It is Fig. A1c.*

lines 316-317. Why is phosphate concentration higher in the Alboran Sea during the LGM? Is this because of enhanced river discharge?

*Yes, the river discharge to the Alboran Sea doubles (triples) in GLAC-1D (ICE-6G) during the LGM. In addition, the maximum of the vertical nutrient profile is shifted to a shallower water depth in the Strait of Gibraltar, which enhances the nutrient availability similar to what we found for the Strait of Sicily (see Fig. 15). The impact of the changed dynamical conditions in the strait on the nutrient distribution is visible for the entire Alboran Sea.*

Lines 395-396, about the agreement between high organic matter accumulation and increased abundance of benthic foraminifera sensitive to food supply during the LGM. A map presenting simulated organic matter content in sediment (Fig. 6) and changes benthic foraminifera abundance will be interesting to show the trend.

*We include information on the relative change of benthic foraminiferal numbers between LGM and PI from Schmiedl et al (2010,2023) in Fig. 6. Data are only available for the eastern basin and show a relative mean increase of ~7 over all locations (ranging between 0.1 to 30), which is of the same order of magnitude ballpark as our simulated relative increase of the deposition flux.*

Line 451, "can not" should be replaced by "cannot".

*Done*

Fig. 1. It will be helpful to add latitudes and longitudes scale since the white line indicates the zonal transect is used Fig. 13 that is *presented with longitudinal scale.*

*We added contour lines of longitude and latitude to Fig.1 for a more convenient comparison of Fig. 13.*

Fig. 4. I am curious to see the result of zonal stream function of PI-Straits at least in supplementary information.

*We added the zonal stream function of PI-Straits (Fig. A1) to the appendix.*

References

Cacho, I., Grimalt, J. O., and Canals, M.: Response of the Western Mediterranean Sea to rapid climatic variability during the last 50,000 years: a molecular biomarker approach, J. MARINE SYST., 33–34, 253-272, 2002.

Essallami, L., Sicre, M. A., Kallel, N., Labeyrie, L., and Siani, G.: Hydrological changes in the Mediterranean Sea over the last 30,000 years, Geochem. Geophys. Geosyst., 8, Q07002, 2007.

Venkatarathnam, K. and Ryan, W. B. F.: Dispersal patterns of clay minerals in the sediments of the eastern Mediterranean Sea, Marine Geology, 11, 261-282, 1971.

---

## Author Comment (AC2)

**Reply to Reviewer #1**

We thank reviewer #1 for the detailed comments in a very friendly manner. We addressed all points with our answer given in Italics following the original comment.

**Comments**

 **Line 23:** I would say "was characterized"…then "is characterized"

*done*

**Line 29:** "Fresher" then "Less saline"

*Changed to fresher*

**Line 32:**  Probably I missed the point in the sentence and thus I'm asking for some additional clarification. As far as I remember the formation of LIW is driven by the salinity. The Authors states that in response to the reduction of the inflow at Gibraltar there is an increase in the salinity in the east Med.. thus I would expect an increase in the LIW formation not a reduction..Could you please explain better this point?

*The short-term effect of a local salinity increase would be an increased formation of LIW. But on longer terms the whole water column and, thus, the stability of the water adjusts to the higher surface salinity. See further discussion under your comment **Paragraph 4.1 and 4.2***

**Line 79:** I would say "coupled" than "combined"

*done*

**Line 93 onwards**: How many biogeochemical tracers considers HAMOCC? How do you represent the phytoplankton in your model? As functional group? Please explain better and provide more information about

*We extended the description of HAMOCC. It now reads:*

*HAMOCC includes a description of the full carbon chemistry, the cycling of nutrients (phosphate, nitrate, silicate, iron) and oxygen, and an extended NPZD-type plankton dynamic with 5 tracers (2 types of phytoplankton, i.e. bulk and nitrogen fixers, one zooplankton type, dissolved and particulate organic matter, the latter called detritus in the following). Phytoplankton growth depends on incoming light at the corresponding depth level, the availability of nutrients and temperature (Paulsen et al, 2017). HAMOCC includes detritus settling and remineralization of dissolved organic matter and detritus (aerobic and anaerobic) as well as production, gravitational sinking, and dissolution of opal and calcite shells (Ilyina et al., 2013, Mauritsen et al., 2019). Shell production is linked to plankton concentration changes with a preference to opal production as long as silicate is available (Ilyina, et al., 2013). Organic matter from primary production is composed of carbon, phosphorus, nitrogen, and oxygen according to a constant stoichiometry ($C:N:P:O_2=122:16:1:-172$, Takahashi, et al., 1985) and iron ($Fe:P=366 \cdot 10^{-6}$, Johnson et al., 1997).*

**Line 128:** How do you consider the period of 1000 year as suitable for the spin up? Did you check the kinetic energy? What about drifts in the tracers?

*From our point of view, a 1000-year spin-up is sufficient for all simulated tracers in the water column. In Figure 1 below, we show the spin-up and the 1000 years of the experiment for the LGM state of GLAC-1D. The transient forcing from the parent simulation shows some variability, but not large trends over the years 22-20 kyrs BP (see Kapsch etal, 2022, doi.org/10.1029/2021GL096767, their Fig. 1b). Physical tracers (T,S) in medHAMOCC are quasi-stationary in the last 250 years of the spin-up phase with little variability induced by the transient forcing. Phosphate shows a higher variability, but no clear trend for the last 400 years of the spin-up. The transient forcing of the 1000 years following the spin-up induces much stronger variability .*

[Figure]

*Fig 1: Variations of decadal means of temperature (upper left), salinity (upper right), and phosphate (lower left) at one location in the Levantine at two depth levels: 815 m (dotted line) and 2315 m (solid line). Shown are the deviations from the means of the years 1001-2000 (grey lines; the absolute mean values are given for each depth level in plots). The spin-up period is indicated by the red arrow in the temperature plot.*

**Line 133:** what about the zenith and albedo?

*We use monthly mean incoming downward short-wave radiation provided by the atmospheric parent model of the MPI-ESM. This includes direct and diffuse components of the downward short-wave radiation and diurnal zenith angle variations. Orbital parameter changes between LGM and PI are also considered in the MPI-ESM simulations. The albedo is set to a constant value in time and space in the physical component of our MedSed model. For the biogeochemistry, we calculate the light availability from the incoming downward short-wave radiation and light absorption due to clear water and due to the simulated phytoplankton concentrations converted to chlorophyll concentration (see Paulsen et al 2017).*

**Line 145 onwards:** The Authors use an open boundary at Gibraltar Strait but they do not impose any speed. Could you please why? Is you E-P+R balance over the Mediterranean sea equal to zero or did you apply any correction to preserve the total volume of the domain? Please explain

*The open boundary is not directly at the Strait of Gibraltar. We include part of the North Atlantic to 10°W (see Fig. 1) to allow for a more or less free development of the dynamics at the entrance of the MedSea. At the western boundary, we include an approx. 80 km wide sponge zone where water properties are relaxed to prescribed values. By setting sea level to zero at the western boundary we guarantee a nearly constant water volume in the model domain and a consistent water flux at the Strait of Gibraltar.*

**Line 185**: "precipitation"

*Corrected*

**Figure 2:** As far as I remember Medar medatlas also provides the uncertainties for each variable and for each depth. I would include them in the figure as error bars to see if they overlap the values simulated by the model. Did you have an explanation for the difference in the oxygen vertical profiles with depth? Respiration processes?

*We included the standard deviation of the Medar medatlas in Fig. 2. Basin averages are calculated from the mean values and their provided standard deviations.*

*Wrt to the difference in oxygen profiles: Oxygen profiles below the euphotic zone are shaped by an interplay between biological processes such as respiration and physical processes such as vertical mixing, the effectiveness of which depends on the water column stability. In view of the impact of biological drivers on oxygen, we see a complementary feature in the nutrient profile with less PO4 in the deep layers of the eastern basin than found in the Medatlas. Both profiles, oxygen and phosphate, point towards a too low export of organic matter and potentially too low primary production. The latter could be a result of a too low nutrient supply via the rivers. However, we base our simulation on literature values of river load (Ludwig etal. 2009) and did not do additional tuning. On the other hand, the Medar climatology may already include an anthropogenic signal from the riverine nutrient supply, which is omitted in our simulation. Wrt the physical drivers: the same figure shows that the temperature profile in the Levantine matches quite well the observations, while the salinity is too low compared to the climatology. This might indicate that we simulate a less stabile water column and thereby overestimate the potential of vertical mixing which, in turn, may lead to a too well-ventilated ocean.*

**Line 203 onwards:** There are some recent results related to the Med ZOC simulated and discussed in Reale et al., 2022. I would include them here. Reale, M., Cossarini, G., Lazzari, P., Lovato, T., Bolzon, G., Masina, S., Solidoro, C., and Salon, S.: Acidification, deoxygenation, and nutrient and biomass declines in a warming Mediterranean Sea, Biogeosciences, 19, 4035–4065, https://doi.org/10.5194/bg-19-4035-2022, 2022.

*Reference included*

**Line 204:** All the listed areas are characterized by deep water formation processes. The Rhodes gyre is the area for the intermediate waters. Please correct

*Our statement refers to the model state. However, even in the real world, intermediate and deep water are formed in the south of the region that we define as Aegean basin (Fig. 1), i.e. the Cretan Sea with the Cretan Intermediate Waters (CIW) and the Cretan Deep Waters (CDW) and in the Rhodes Gyre for the Levantine Intermediate Waters (LIW) and the Levantine Deep Waters (LDW) according to Simoncelli and Pinardi (3.4. Water mass*

*formation processes in the Mediterranean Sea over the past 30 years, in Schuckmann et al.,2018, Copernicus Marine Service Ocean State Report, Journal of Operational Oceanography,* [https://doi.org/10.1080/1755876X.2018.1489208).](https://doi.org/10.1080/1755876X.2018.1489208)  *We did not change the sentence.*

**Line 209 onwards:** I would provide more quantitative information here about the differences between simulated and observed values. The Authors could use for example for the NPP the table 4 in Reale et al., 2020 Reale, M., Giorgi, F., Solidoro, C., Di Biagio, V., Di Sante, F., & Mariotti, L., et al. (2020). The regional Earth system Model RegCM-ES: Evaluation of the Mediterranean climate and marine biogeochemistry.

*We added a table in Appendix A which provides the annual mean net primary production of GLAC-1D and ICE-6G for the PI and a selection of literature values. We added in the main text: "Compared to satellite derived estimates from Uitz et al (2012), our simulated NPP is rather on the low side, but fits well within the range of other modelling studies (Table A1). However, the absolute value of the NPP is not so relevant for our study. The amount of exported nutrient from the surface layers below the euphotic zone is more important, as it shapes the nutrient profile."*

**Line 220 onwards:** 10-20 % is not slightly higher. Please correct

*We agree and deleted the word "slightly".*

**Figure 7 and Paragraph 4** The authors talk about anomalies and biases. As far as I see they are differences. I would call them like that. I would test if these differences are statistically significant and I would mark the map with a dot where this happens.

*The word "anomaly" is the common nomenclature to refer to the difference between temporal mean tracer distributions e.g. the LGM and the PI (see e.g. Kageyama et al., The PMIP4 Last Glacial Maximum experiments: preliminary results and comparison with the PMIP3 simulations, Clim. Past, 17, 1065–1089, https://doi.org/10.5194/cp-17-1065-2021, 2021).*

*Regarding the term "bias":*

1) *We use the term in connection of the comparison of model results with observations or reanalysis data (e.g. the description of the bias correction in line 147). All models exhibit spatial deviations from "reality" which result, among other things, from the temporal and spatial resolution or the parameterization of the processes.*
2) *Biases are also mentioned when describing paleoceanographic tools, such as the alkenone unsaturation ratio to estimate the annual mean sea surface temperature. We follow here the nomenclature of the authors (e.g. Conte et al., 2006, Global temperature calibration of the alkenone unsaturation index (U37 K03) in surface waters and comparison with surface sediments, Geochem. Geophys. Geosyst., 7, Q02005, doi:10.1029/2005GC001054)*

*There is no use for carrying out a statistical analysis. We averaged our data over 1000 years and the obtained LGM and PI mean states are very different. Changes are everywhere much larger than e.g. the temporal variability of the PI state. We expect insignificant signals only in the few areas where is anomaly is close to zero.*

**Line 262-264:** Could you explain better this point since it is not very clear

*To understand the pronounced increase in SSS that is found in the Ionian Sea, we have to look at circulation changes in more detail. We added here only a note on the later discussion on pycnocline changes.*

**Paragraph 4.1 and 4.2** as far as I understand from the results the SST gets colder and thus I would expect a decrease in the vertical stability and increase in deep water formation processes. Could you please explain better why happens the opposite?

*As stated above (see our answer to your comment on **line32**), a short-term change in sea surface properties, either a colder SST or increased SSS, will, of course, lead to enhanced deep-water formation. But if we run the model persistently with colder SST and higher SSS as provided by the LGM conditions, starting e.g. from PI conditions for T and S in the water column, the enhanced deep-water formation fills up the entire deep basins and increases the water column stability. It will take a long time for the surface signal to change the entire MedSea conditions so that dense enough water can form again to overcome the stratification.*

*However, the dominant effect is related to the reduction and shallowing of the inflow at the Strait of Gibraltar which alone causes a shallowing of the pycnocline depth (Rohling, 1991). Rohling also demonstrates that the reduced inflow invokes an increase in the salinity contrast between the upper und the deeper layers. The pycnocline depth change due to a reduced inflow is shown by our sensitivity study PI-Straits (Fig. 10b in the paper). Figure 2 shows the changes of the surface salinity in PI-Straits due to the reduced exchange at the Straits of Gibraltar and the corresponding maximum mixed layer depth (MLD) over 2000 years of one location in the eastern basin. As mentioned in the paper, the slow-down of the ZOC increases the residence time of the water in the eastern basin and leads to the SSS increase. This SSS increase invokes deep MLDs in the first 400 years during the spin-up. After this time of circulation adjustment, MLDs in PI-Straits are lower than in the PI experiment which indicates the increased stratification.*

[Figure]

[Figure]

*Fig 2: Time series of maximal annual mixed layer depth (upper panel) and sea surface salinity (lower panel) for a point south of the Peloponnese for PI of GLAC-1D (blue) and PI-Straits (orange) over the 1000 years spin-up and 1000 years used for analysis. Salinity data are 10yr averages, mixed layer depths are the seasonal maximum of decadal data.*

---

## Author Response (AR1)

We revised the manuscript including all modifications made in response to the reviewers' comments. We have also updated the figures and added two requested figures and a table to the appendix.

---

## Author Response (AR2)

We revised the final version of the manuscript according to the latest editorial suggestions.

Thank you for the paper handling.

Best regards, Katharina Six